Letter

# Autoantibodies against chemokines post-SARS-CoV-2 infection correlate with disease course

Jonathan Muri [1,23], Valentina Cecchinato [1,23], Andrea Cavalli [1,2 ✉],
Akanksha A. Shanbhag[1], Milos Matkovic[1], Maira Biggiogero[3],
Pier Andrea Maida[3], Jacques Moritz[1], Chiara Toscano[1], Elaheh Ghovehoud [1],
Raffaello Furlan[4,5], Franca Barbic [4,5], Antonio Voza[4,6], Guendalina De Nadai[7],
Carlo Cervia [8], Yves Zurbuchen [8], Patrick Taeschler [8], Lilly A. Murray [9],
Gabriela Danelon-Sargenti[1], Simone Moro[1], Tao Gong[1], Pietro Piffaretti[1],
Filippo Bianchini[1], Virginia Crivelli[1], Lucie Podešvová[1], Mattia Pedotti[1],
David Jarrossay[1], Jacopo Sgrignani[1], Sylvia Thelen[1], Mario Uhr[10],
Enos Bernasconi [11,12], Andri Rauch [13], Antonio Manzo[14], Adrian Ciurea [15],
Marco B. L. Rocchi[16], Luca Varani [1], Bernhard Moser[17], Barbara Bottazzi [18],
Marcus Thelen [1], Brian A. Fallon[9,19], Onur Boyman [8,20],
Alberto Mantovani [4,18,21], Christian Garzoni[22], Alessandra Franzetti-Pellanda[3],
Mariagrazia Uguccioni [1,4,24 ✉] & Davide F. Robbiani [1,24 ✉]

Infection with severe acute respiratory syndrome coronavirus 2 associates with diverse symptoms, which can persist for months. While antiviral antibodies are protective, those targeting interferons and other immune factors are associated with adverse coronavirus disease 2019 (COVID-19) outcomes. Here we discovered that antibodies against specific chemokines were omnipresent post-COVID-19, were associated with favorable disease outcome and negatively correlated with the development of long COVID at 1 yr post-infection. Chemokine antibodies were also present in HIV-1 infection and autoimmune disorders, but they targeted different chemokines compared with COVID-19. Monoclonal antibodies derived from COVID-19 convalescents that bound to the chemokine N-loop impaired cell migration. Given the role of chemokines in orchestrating immune cell trafficking, naturally arising chemokine antibodies may modulate the inflammatory response and thus bear therapeutic potential.

The spectrum of disease manifestations upon infection with severe acute respiratory syndrome coronavirus 2 (SARS-CoV-2) is broad and COVID-19 convalescent individuals often lament protracted symptoms over months, a condition referred to as long COVID or PASC (post-acute sequelae of COVID)[1–5]. Some evidence points to a role for immune dysregulation and autoimmunity as contributors to long COVID, although

virus persistence has also been proposed[6–8]. Overall, however, there is little understanding of the biology underlying long COVID and the reasons behind the differences in COVID-19 manifestation.

Chemokines are chemotactic cytokines that mediate leukocyte trafficking and activity[9]. In addition to elevated levels of pro-inflammatory cytokines, acute COVID-19 is characterized by higher

expression of certain chemokines[10–16]. Accordingly, chemokines recruit neutrophils and monocytes to sites of infection, where they play a key role in the pathophysiology of COVID-19 by sustaining inflammation and causing collateral tissue damage and fibrosis[11,15,17,18]. Autoantibodies to type I interferon (IFN) and other molecules are observed in COVID-19 and are generally associated with adverse outcome[19–25]. However, no study comprehensively investigated chemokine antibodies, nor their persistence over time.

Here we devised a peptide-based strategy to measure antibodies that bind to a functional region of each of the 43 human chemokines. By examining three independent COVID-19 cohorts, we found that the presence of autoantibodies against specific chemokines helped identify convalescent individuals with favorable acute and long COVID disease course. Monoclonal antibodies targeting chemokines that were derived from these individuals blocked leukocyte migration and thus may be beneficial through modulation of the inflammatory response.

To evaluate chemokine antibodies after COVID-19, we obtained plasma samples from a cohort of COVID-19 convalescents ($n = 71$) at month 6 (on average) post disease onset (hereafter, the Lugano cohort; Supplementary Tables 1 and 2). Plasma samples from COVID-19 uninfected individuals, confirmed by negative serologic test, were used as healthy controls ($n = 23$). Because the N-terminal loop (N-loop) of chemokines is required for receptor binding, we investigated whether biologically active chemokine antibodies targeted this region, whose sequence is specific for each but two of the 43 human chemokines (Extended Data Fig. 1a)[26]. We designed peptides corresponding to the N-loop of each chemokine for use in ELISA assays (Supplementary Table 3), measured the amount of peptide-specific IgG antibody in serial plasma dilutions and plotted the signal as a heatmap (Fig. 1a and Extended Data Fig. 1b). Analysis of all parameters by nonlinear dimensionality reduction with $t$-distributed stochastic neighbor embedding ($t$-SNE) indicated a clear separation between healthy controls and COVID-19 convalescents (Extended Data Fig. 2a). Some COVID-19 convalescent plasma indicated high levels of IgGs to certain chemokines (for example, CCL8, CXCL13 and CXCL16) compared with healthy controls. For these chemokines, antibody levels to the N-loop significantly correlated with those against the C-terminal region of the same chemokine, suggesting that, when present, antibodies formed against multiple chemokine epitopes (Fig. 1a and Extended Data Fig. 2b). When considering antibodies against each chemokine individually, a significant difference in reactivity, shown as ratio over healthy controls, was observed for peptides corresponding to 23 of the 43 chemokines (Extended Data Fig. 2c). Antibodies to the three chemokines with $P < 10^{-4}$ (CCL19, CCL22 and CXCL17) clustered together and by themselves were sufficient to correctly assign healthy controls and COVID-19 convalescents with high accuracy (96.8%; Fig. 1a–c and Extended Data Fig. 2c,d), so they were defined as a 'COVID-19 signature'. This signature was validated in two independent COVID-19 cohorts.

The Milan cohort was sampled during the acute phase (day 8, on average) and at month 7 from disease onset ($n = 44$; 90.5% and 89.5% accuracy, respectively; Extended Data Fig. 2e and Supplementary Tables 2 and 4), and the Zurich cohort was evaluated at month 13 from disease onset ($n = 104$; 92.9% accuracy; Extended Data Fig. 2f and Supplementary Tables 2 and 4). Thus, COVID-19 was associated with a specific pattern of autoantibodies against chemokines.

To examine the relationship between chemokine antibodies and other serologic features of the Lugano cohort, we used ELISA and a pseudovirus-based neutralization assay to measure the binding and neutralizing capacity of antibodies against SARS-CoV-2 (ref. [27]). In agreement with previous studies[27], IgG binding to the Spike receptor binding domain (RBD) of SARS-CoV-2 and plasma half-maximal SARS-CoV-2 neutralizing titers ($NT_{50}$) were variable (Fig. 1a), but positively correlated with each other and with age (Extended Data Fig. 3a–c)[27]. In contrast, there was no correlation of $NT_{50}$ or RBD IgGs with the levels of COVID-19 signature chemokine antibodies (against CCL19, CCL22 and CXCL17), or with the sum of all chemokine IgG reactivities (Extended Data Fig. 3d). A weak negative correlation between age and the sum of all chemokine IgG reactivities was observed (Extended Data Fig. 3d), but there were no significant differences in the levels of the COVID-19 signature chemokine antibodies between males and females (Extended Data Fig. 3e). These observations indicated that, after COVID-19, autoantibodies against specific chemokines were not correlated with the antibodies against SARS-CoV-2.

To document the temporal evolution of chemokine antibodies following COVID-19, we compared the reactivities of plasma collected from the Lugano cohort at around month 6 and month 12 post-symptom onset side-by-side (Extended Data Fig. 4a and Supplementary Table 2). In agreement with earlier findings[28], RBD antibodies significantly decreased in unvaccinated COVID-19 convalescents, while they increased in those receiving at least one dose of messenger RNA-based COVID-19 vaccine (Extended Data Fig. 4b and Supplementary Table 2). Conversely, and regardless of vaccination status, antibodies to CCL19, CCL8, CCL13, CCL16, CXCL7 and CX3CL1 significantly increased, those to CXCL17 remained generally stable and those to CCL22 followed variable kinetics at month 12 compared with month 6 in the COVID-19 convalescents (Fig. 1d and Extended Data Fig. 4c). To further investigate the kinetics of COVID-19 signature chemokine antibodies, we analyzed COVID-19 convalescents from the Lugano cohort for which acute samples were also available ($n = 12$; Extended Data Fig. 4d and Supplementary Table 2). During acute COVID-19, IgG antibodies to CCL19, but not to CCL22 or CXCL17, were already higher than in healthy controls, and continued to increase until month 12 (Extended Data Fig. 4e). Similarly, in the Milan cohort, autoantibodies against CCL19, CCL22 and CXCL17 were higher during the acute phase than in healthy controls (Extended Data Fig. 2e). In contrast to natural infection, no significant change in antibody reactivity to any of the chemokines

**Fig. 1 | Distinct patterns of chemokine antibodies in COVID-19 convalescents with different severity of acute disease. a**, Heatmap representing plasma IgG binding to 42 peptides comprising the N-loop of all 43 human chemokines, as determined by ELISA in healthy controls (Controls) and COVID-19 convalescents (COVID-19) of the Lugano cohort at month 6. Samples are ranked according to the level of SARS-CoV-2 RBD reactivity. Chemokine IgGs are ordered by unsupervised clustering analysis of ELISA signal. SARS-CoV-2 pseudovirus neutralizing activity ($NT_{50}$) and IgG binding to peptides corresponding to negative control, IFN-α2 and SARS-CoV-2 nucleocapsid protein (N) are shown. **b**, AUC of ELISA showing IgG antibodies to CCL19, CCL22 and CXCL17 (COVID-19 signature) in healthy controls and COVID-19 convalescents at month 6. Two-tailed Mann–Whitney $U$-test. **c**, Logistic regression analysis showing the assignment of COVID-19 convalescents and healthy controls based on CCL19, CCL22 and CXCL17 antibodies at month 6. **d**, AUC of ELISA showing CCL19, CCL22 and CXCL17 antibodies at months 6 and 12 in COVID-19 convalescents ($n = 63$). Wilcoxon two-tailed signed-rank test. **e**, Chemokine antibodies in previously hospitalized and outpatient COVID-19 convalescents at month 6, shown as ratio over healthy controls. Circle size indicates significance; colors show the $\log_2$ fold-change increase (red) or decrease (blue), shown as ratio over healthy controls. Kruskal–Wallis test followed by Dunn's multiple comparison test. **f**, Cumulative AUC of ELISA signal of the IgGs against the 42 chemokine N-loops in healthy controls and previously hospitalized and outpatient COVID-19 convalescents at month 6. Kruskal–Wallis test followed by Dunn's multiple comparison test. **g**, $t$-SNE distribution of previously hospitalized and outpatient COVID-19 convalescents at month 6, as determined with the 42 datasets combined. **h**, Logistic regression analysis showing the assignment of previously hospitalized and outpatient COVID-19 convalescents based on CXCL5, CXCL8 and CCL25 antibodies (COVID-19 hospitalization signature) at month 6. In **b** and **f**, horizontal bars indicate median values. In **a–h**, AUC values are the average from two independent experiments. Healthy controls ($n = 23$) in **a**, **b**, **c**, **e** and **f**; COVID-19 convalescents ($n = 71$) in **a**, **b** and **c**, of which previously hospitalized ($n = 50$) and outpatient ($n = 21$) in **e–h**. Ab, antibody; AUC, area under the curve; FC, fold-change; ID, identity; m, months; NS, not significant.

was observed upon COVID-19 mRNA vaccination of SARS-CoV-2 naïve individuals at around month 4 post-vaccination (130 d on average; n = 16; Extended Data Fig. 4f and Supplementary Table 2). Thus, unlike the antibodies to SARS-CoV-2 RBD, which decreased over time, the levels of some chemokine antibodies that were present during acute COVID-19 increased over 1 yr of observation.

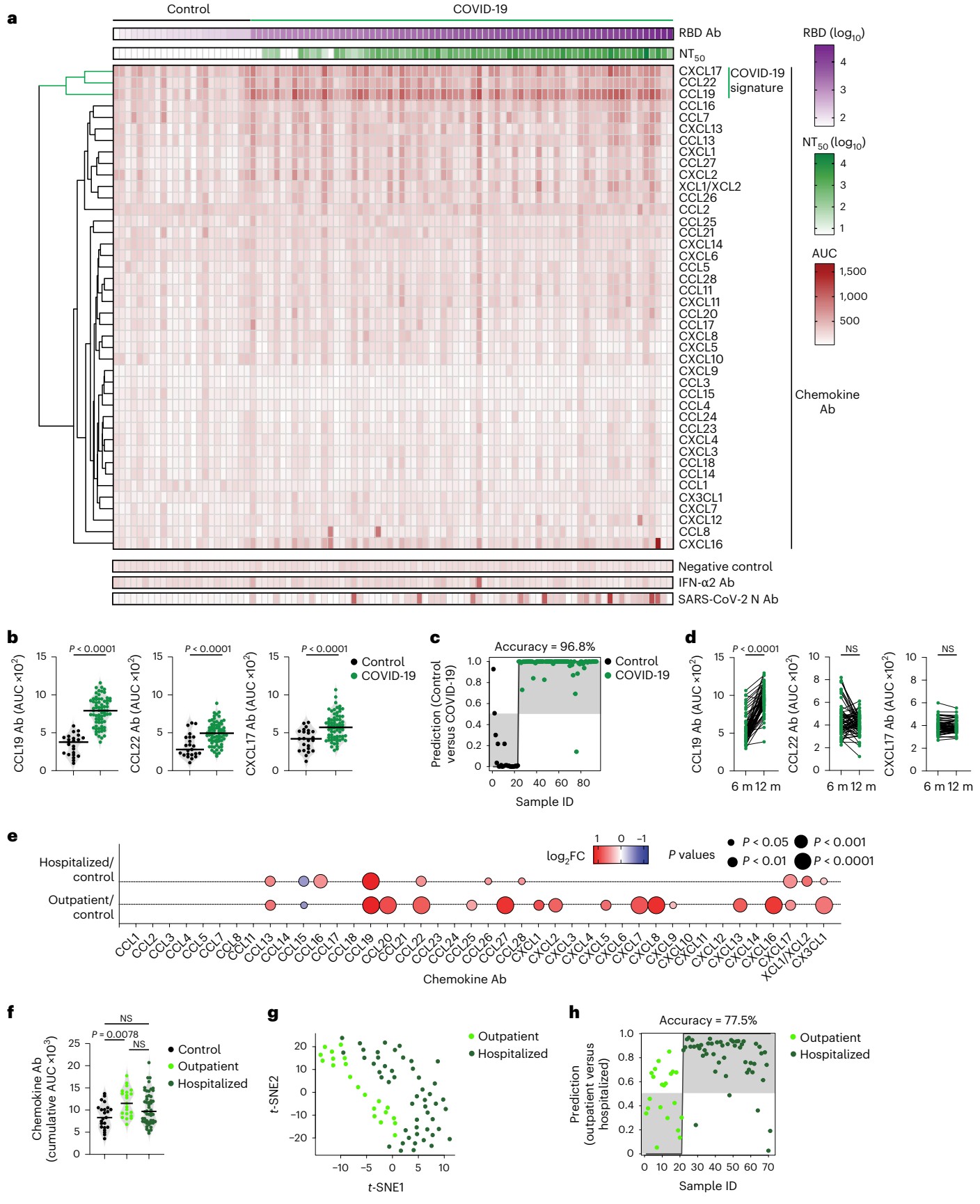

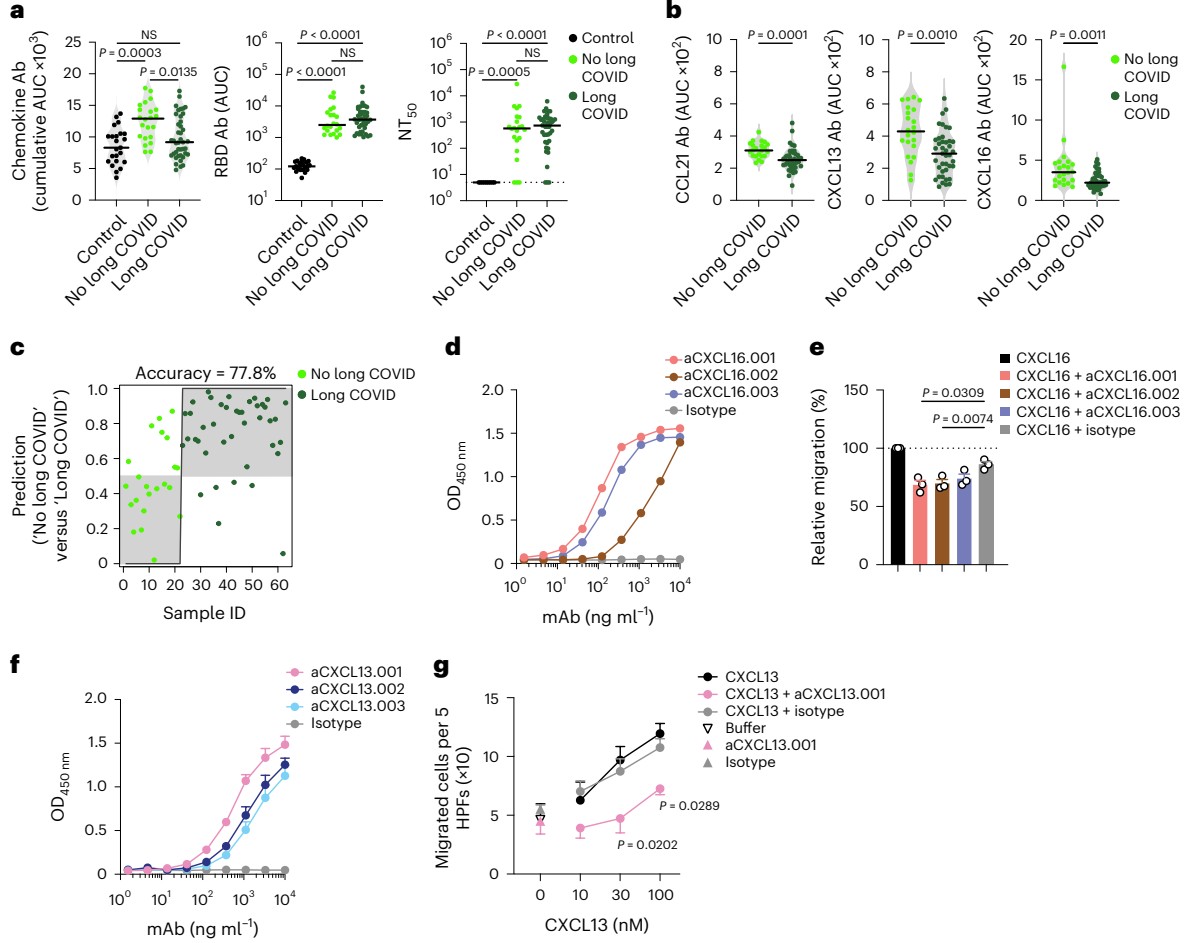

**Fig. 2 | Autoantibodies against specific chemokines in COVID-19 convalescents without persistent symptoms at month 12. a**, Chemokine IgG (cumulative AUC of ELISA; left), RBD IgG (middle) and NT50 (right) values in healthy controls (Controls) and COVID-19 convalescents (COVID-19) of the Lugano cohort at month 6 grouped as long COVID and no long COVID at month 12. Kruskal–Wallis test followed by Dunn's multiple comparison test. **b**, AUC of ELISA showing CCL21, CXCL13 and CXCL16 antibodies (long COVID signature) at month 6 in COVID-19 convalescents defined as long COVID and no long COVID at month 12. Two-tailed Mann–Whitney $U$-tests. **c**, Logistic regression analysis showing the assignment of COVID-19 convalescents as long COVID and no long COVID at month 12, based on CCL21, CXCL13 and CXCL16 antibodies at month 6. **d**, ELISA showing CXCL16 antibodies binding to the CXCL16 N-loop peptide. Average of two independent experiments. **e**, Chemotaxis showing relative migration of the 300.19 preB cell line uniquely expressing CXCR6 in a CXCL16 gradient (1 nM). Mean + s.e.m. of three independent experiments. Paired, two-tailed Student's $t$-test. **f**, ELISA showing CXCL13 antibodies binding to the CXCL13 N-loop peptide. Average of four independent experiments (mean + s.e.m.). **g**, Chemotaxis of primary CD19+ human B cells isolated from buffy coats in a CXCL13 gradient in the presence of the aCXCL13.001 antibody or isotype control. Mean ± s.e.m. of migrated cells in five high-power fields (HPFs). Average of three independent experiments with cells from different donors. Two-way repeated measures ANOVA followed by Šídák's multiple comparisons test. In **a** and **b**, horizontal bars indicate median values and data are shown as average AUC from two independent experiments. Healthy controls ($n$ = 23) in **a**; COVID-19 convalescents without ($n$ = 22) or with ($n$ = 41) long COVID at month 12 in **a**, **b** and **c**. OD, optical density.

The presence of autoantibodies in a portion of hospitalized patients with COVID-19 is linked to severe illness[20,21,24,25]. No significant difference in age distribution was observed between the previously hospitalized ($n$ = 50) and outpatient ($n$ = 21) COVID-19 convalescents in the Lugano cohort (60 ± 14 and 57 ± 15 yr, respectively), while a higher proportion of males was observed in the formerly hospitalized, but not outpatient, group (60% and 38.1%, respectively; Supplementary Table 2). When the most significant differences in autoantibody levels at month 6 were considered ($P < 10^{-4}$), only the antibodies against CCL19 were higher in previously hospitalized COVID-19 convalescents compared with healthy controls, while antibodies against 8 chemokines (CXCL8, CCL22, CXCL16, CCL27, CXCL7, CCL20, CX3CL1 and CCL19) were increased in outpatient COVID-19 convalescents compared with healthy controls (Fig. 1e and Extended Data Fig. 5a). Only the outpatient COVID-19 convalescents displayed significantly higher cumulative IgG reactivity against the chemokines compared with healthy controls (Fig. 1f).

Thus, a broader pattern and higher overall amounts of autoantibodies against chemokines were observed at month 6 in COVID-19 convalescents who were outpatients during the acute phase of the disease.

Direct comparison by $t$-SNE analysis of all chemokine antibodies at month 6 separated COVID-19 convalescents in the Lugano cohort that were previously hospitalized from outpatients (Fig. 1g). Antibodies against CXCL5, CXCL8 and CCL25 were all lower in previously hospitalized compared with outpatient COVID-19 convalescents (Extended Data Fig. 5a,b), and this was not linked to the therapy received during hospitalization (Supplementary Table 2). The combination of antibody values against CXCL5, CXCL8 and CCL25 at month 6 alone could correctly assign formerly hospitalized and outpatient COVID-19 convalescents with an accuracy of 77.5% ('COVID-19 hospitalization signature'; Fig. 1h). Similar findings were obtained in the Milan (85.0% and 84.1% accuracy at acute and month 7, respectively; Extended Data Fig. 5c) and Zurich cohorts (73.1% accuracy at month 13; Extended Data Fig. 5d).

In the COVID-19 hospitalization signature, antibodies to CXCL5 and CXCL8 were negatively correlated with RBD IgG and age, but not with sex (Extended Data Fig. 6a,b). Consistent with previous work[27], both RBD IgG and $NT_{50}$ values were significantly higher in previously hospitalized compared with outpatient COVID-19 convalescents and in males compared with females of both hospitalized and outpatient groups (Extended Data Fig. 6c)[27]. Thus, the chemokine antibody signature that distinguished healthy controls from COVID-19 convalescents (autoantibodies to CCL19, CCL22 and CXCL17) was different from the signature associated with COVID-19 severity (autoantibodies to CXCL5, CXCL8 and CCL25).

A fraction of individuals who recover from COVID-19 experience long-term sequelae[1,3,4]. To determine whether a specific pattern of chemokine antibodies at month 6 was predictive of long COVID, we used a questionnaire to collect information on self-reported symptoms (for example, pulmonary, systemic, neurological and psychiatric) at month 12 from the Lugano cohort (Supplementary Table 5). 65.1% of all participants reported the persistence of at least one symptom related to COVID-19. Among these, the average number of long-term symptoms was 3.3, and they were more frequent in formerly hospitalized (72.7%) compared with outpatient (47.4%) COVID-19 convalescents (Extended Data Fig. 7a and Supplementary Tables 2 and 5). No differences in age or sex distribution or time from disease onset to month 12 visit were observed between COVID-19 convalescents with protracted symptoms at month 12 (long COVID) or without (no long COVID) (Extended Data Fig. 7b).

COVID-19 convalescents in the Lugano cohort with long COVID, particularly outpatients and females, showed significantly lower cumulative levels of chemokine antibodies compared with those with no long COVID (Fig. 2a and Extended Data Fig. 7c,d). In contrast, RBD IgG and $NT_{50}$ values were comparable between long COVID and no long COVID groups (Fig. 2a). The cumulative amount of chemokine antibodies did not correlate with the number of symptoms (Extended Data Fig. 7e). IgG antibodies against CCL21, CXCL13 and CXCL16 at month 6 distinguished long COVID from no long COVID groups with high significance and were defined as a 'long COVID signature' (Fig. 2b and Extended Data Fig. 7f). Logistic regression analysis using the antibody values for CCL21, CXCL13 and CXCL16 alone predicted the absence of persistent symptoms with 77.8% accuracy (Fig. 2c). Similarly, analysis of the Zurich cohort at month 13 showed 72.1% accuracy of association with lack of long COVID, even though in that cohort only CCL21 antibodies were significantly different between long COVID and no long COVID groups (Extended Data Fig. 7g). These results indicated that specific patterns of chemokine antibodies at month 6 were associated with the longer-term persistence of symptoms after COVID-19.

Because chemokine antibodies to CXCL13 and CXCL16 were associated with decreased likelihood of long COVID, we next derived corresponding memory B cell antibodies from available peripheral blood mononuclear cell (PBMC) samples (Supplementary Table 6). Three N-loop binding monoclonal antibodies were obtained for CXCL16, which blocked migration of the 300.19 preB cell line expressing the CXCL16 cognate receptor CXCR6 (Fig. 2d,e, Extended Data Fig. 8a,b and Supplementary Tables 6 and 7). Similarly, three CXCL13 N-loop antibodies bound in ELISA and inhibited chemotaxis of primary CD19+ human B cells (Fig. 2f,g, Extended Data Fig. 8c and Supplementary Tables 6 and 7). By the same approach, we discovered chemotaxis-blocking antibodies specific for CCL8 and CCL20 (Extended Data Fig. 8d–j and

Supplementary Tables 6 and 7). Consistent with these results, polyclonal plasma IgG from COVID-19 convalescents effectively blocked chemotaxis at concentrations 50 times lower than those found in human serum[29] (Extended Data Fig. 8k). Thus, antibodies from COVID-19 convalescents that bound to the N-loop of chemokines were biologically active.

To test the correlation between autoantibodies and their corresponding antigens, we measured plasma chemokine levels in COVID-19 convalescents from the Milan (acute and month 7) and Lugano (acute, month 6 and 12) cohorts. In agreement with earlier reports[10–16], the plasma amount of several chemokines was significantly elevated in the Milan cohort both during acute disease (CCL2, CCL3, CCL4, CCL19, CCL21, CCL22, CCL25, CXCL2, CXCL8, CXCL9, CXCL10, CXCL13 and CXCL16) and at month 7 (CCL19, CCL21, CCL22, CXCL2, CXCL8, CXCL10, CXCL13 and CXCL16) in COVID-19 convalescents compared with healthy controls (Fig. 3a). Similar results were observed with the 12 individuals from the Lugano cohort for whom acute samples were available (Fig. 3a). Of note, CXCL5, CXCL8 and CCL25, which correspond to the chemokine antibodies representing the COVID-19 hospitalization signature, were not significantly different between mild and severe hospitalized COVID-19 convalescents in the Milan cohort (Fig. 3b).

We observed no correlation between the amounts of chemokines and the amounts of corresponding autoantibodies in the acute phase or at month 7 post-infection in the Milan cohort (Fig. 3c). We also did not detect an increase in CCL3 and CCL4 antibodies in COVID-19 convalescents compared with healthy controls in the Lugano cohort at month 6, although the amount of corresponding chemokines was elevated in their plasma (Fig. 3a and Extended Data Fig. 5a). As such, even though various chemokines rapidly increased and persisted in plasma for at least 6 months post-infection, their level did not correlate with the amount of corresponding autoantibodies in the circulation.

Next, we measured chemokine antibodies in plasma from patients chronically infected with HIV-1 ($n = 24$) or *Borrelia* (Lyme disease, $n = 27$); and from patients with ankylosing spondylitis (AS, $n = 13$), rheumatoid arthritis (RA, $n = 13$) or Sjögren syndrome (SjS, $n = 13$); or healthy controls ($n = 23$ for HIV-1, AS, RA, SjS; $n = 30$ for *Borrelia*) (Fig. 4 and Supplementary Table 2). Antibodies against 14 chemokines (CCL2, CCL3, CCL4, CCL5, CCL20, CCL21, CCL22, CCL23, CCL27, CCL28, CXCL7, CXCL8, CXCL9 and CXCL12), but not against CCL19, were significantly increased ($P < 10^{-4}$) in HIV-1-infected individuals compared with healthy controls (Fig. 4a,b and Extended Data Fig. 9a). Antibodies against four chemokines (CCL4, CCL19, CCL25 and CXCL9) were increased in AS, RA and SjS compared with healthy controls (Fig. 4a,b and Extended Data Fig. 9a,b). Plasma from *Borrelia*-infected individuals was indistinguishable from the healthy controls, except for elevated CXCL14 antibodies in the acute phase (Extended Data Fig. 10a and Supplementary Table 2). Unsupervised clustering analysis (Extended Data Fig. 10b) or t-SNE analysis (Fig. 4c) of all chemokine antibody values correctly categorized all COVID-19 and HIV-1 samples with 100% accuracy, while AS, RA and SjS clustered with each other. Thus, patterns of autoantibodies against chemokines not only distinguished different COVID-19 trajectories, but also characterized other infections and autoimmune disorders.

Here we showed that autoantibodies against chemokines were omnipresent after SARS-CoV-2 infection, and that high expression of specific chemokine antibodies was associated with favorable disease

**Fig. 3 | Concentration of plasma chemokines during acute COVID-19 and in convalescence. a**, Plasma chemokine levels in the Milan ($n = 44$; acute and month 7) and Lugano ($n = 12$; acute, months 6 and 12) cohorts compared with healthy controls (Controls, $n = 11$). Horizontal bars indicate median values. Kruskal–Wallis test followed by Dunn's multiple comparison test over healthy controls. **b**, Concentration of CXCL5, CXCL8 and CCL25 in healthy controls ($n = 11$) versus acute ($n = 12$, mild hospitalized; $n = 26$, severe hospitalized) and month 7 ($n = 13$, mild hospitalized; $n = 31$, severe hospitalized) COVID-19 convalescents in the Milan cohort. Horizontal bars indicate median values. Kruskal–Wallis test followed by Dunn's multiple comparison test. **c**, Correlation between levels of chemokine (acute) and autoantibody (acute or month 7) by two-tailed Pearson correlation analysis in mild hospitalized ($n = 12$) and severe hospitalized ($n = 26$) COVID-19 convalescents in the Milan cohort.

outcomes. These observations, in three independent cohorts, contrast with previous reports that connected autoantibodies to severe disease in COVID-19 and other infections[19–22,25,30–32].

Several chemokines are detected in high amounts in bronchoalveolar and other fluids during COVID-19, fueling a pro-inflammatory environment in the lungs, which likely contributes to critical illness and

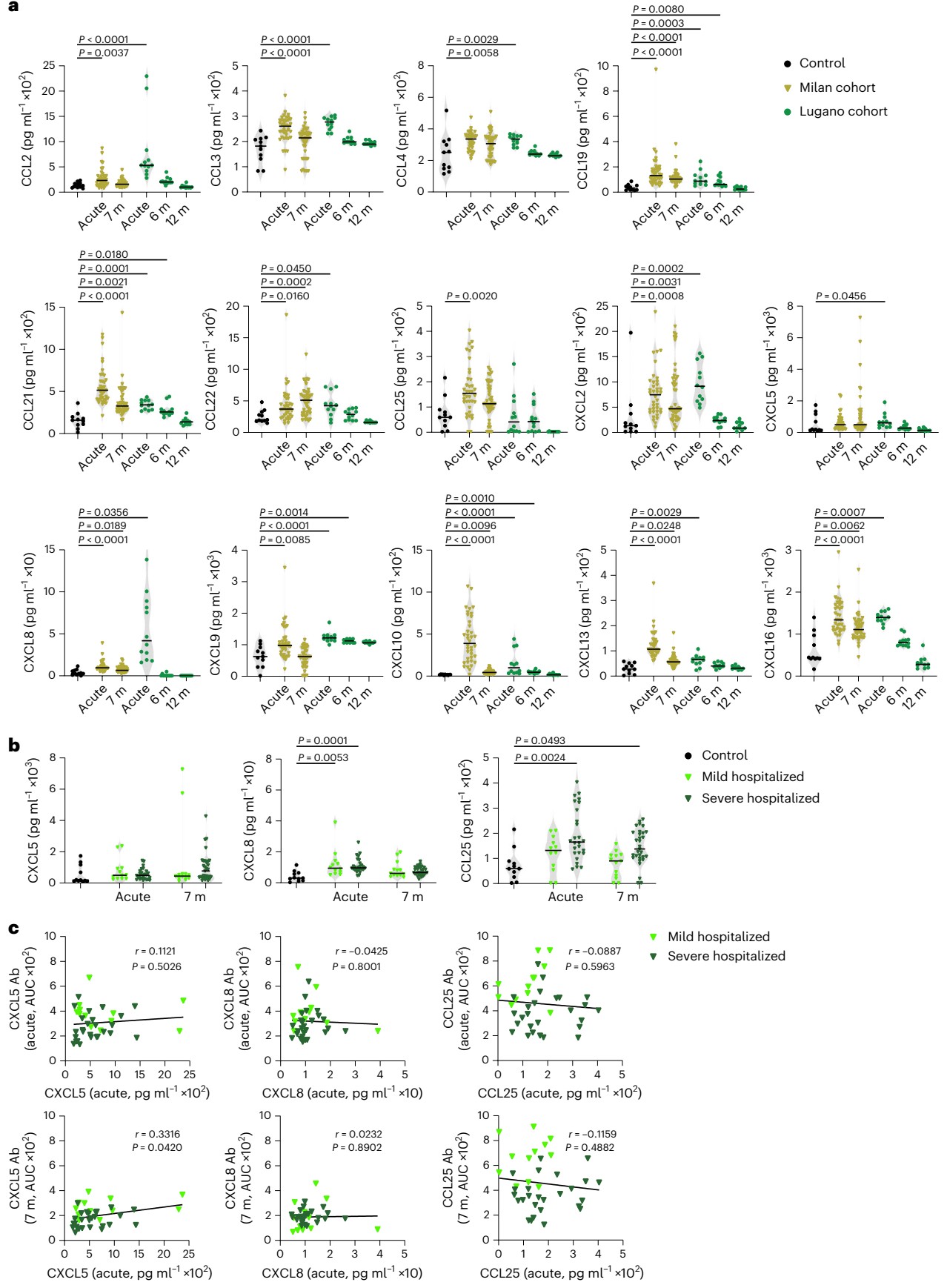

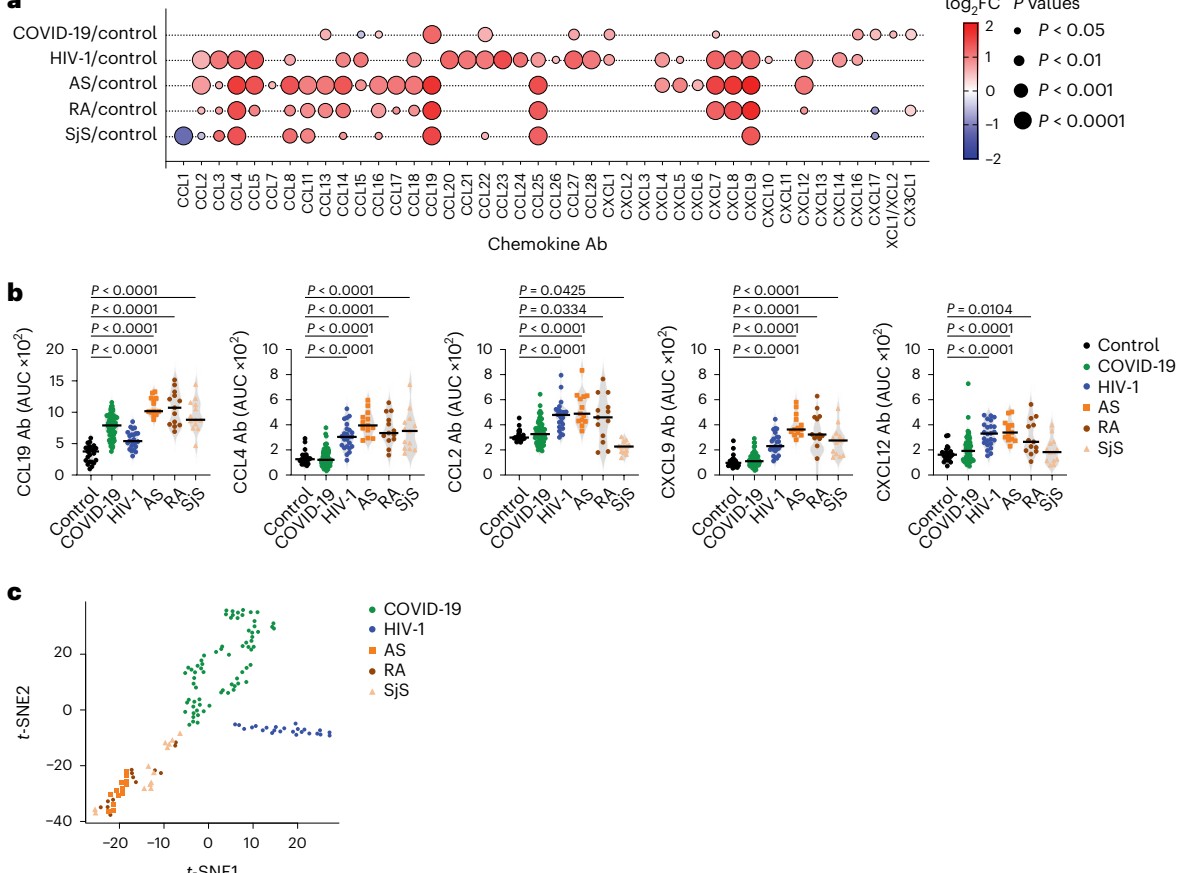

**Fig. 4 | Patterns of chemokine antibodies in COVID-19, HIV-1 and autoimmune diseases. a**, Autoantibodies against specific chemokines in COVID-19 convalescents (COVID-19, month 6, Lugano cohort) and in patients with HIV-1, AS, RA and SjS, shown as ratio over healthy controls (Control). Circle size indicates significance; colors show the $\log_2$ fold-change increase (red) or decrease (blue), shown as ratio over healthy controls. Kruskal–Wallis test followed by Dunn's multiple comparison test. **b**, AUC of ELISA showing IgG antibodies to CCL19, CCL4, CCL2, CXCL9 and CXCL12 in the same groups as in **a**. Horizontal bars indicate median values. Average AUC from two independent experiments. Kruskal–Wallis test followed by Dunn's multiple comparison test over rank of the healthy control group. **c**, t-SNE distribution of COVID-19 convalescents (month 6) and patients with HIV-1, AS, RA and SjS, as determined with the 42 datasets combined. In **a**–**c**, healthy controls ($n = 23$), COVID-19 ($n = 71$), HIV-1 ($n = 24$), AS ($n = 13$), RA ($n = 13$) and SjS ($n = 13$).

hospitalization[10–14]. We found autoantibodies against CXCL5, CXCL8 and CCL25 in COVID-19, but there was no correlation with the amount of the corresponding chemokines in plasma. Because these chemokines attract neutrophils and other cell types that promote inflammation and tissue remodeling, the presence of the corresponding autoantibodies suggests protection through dampening of the damaging inflammatory response associated with severe COVID-19.

Autoantibodies to CCL21, CXCL13 and CXCL16 were increased in recovered individuals compared with those with long COVID 1 yr post-infection. These chemokines are important for tissue trafficking and activation of T and B lymphocytes. It is possible that their respective autoantibodies positively impact the long-term outcome of COVID-19 by antagonizing or otherwise modulating the activation, recruitment and retention of these cell types. Persistent immune responses have been proposed as a mechanism for long COVID, and chemokines have been implicated in its pathogenesis[1,33].

Infection can trigger antibody polyreactivity and autoimmunity which are generally deleterious[34–36]. Because chemokine antibodies are present in plasma after COVID-19 at concentrations able to impair cellular migration, the variety and amount of chemokine antibodies that are present or induced upon infection in each individual may positively modulate the quality and strength of the inflammatory response, which in turn would impact disease manifestation, severity and long COVID. Further studies are needed to determine whether agents that target

the chemokine system could impact positively on the early inflammatory phase of COVID-19 and reduce the development of long COVID.

## Online content

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

[1]Institute for Research in Biomedicine, Università della Svizzera italiana, Bellinzona, Switzerland. [2]Swiss Institute of Bioinformatics, Lausanne, Switzerland. [3]Clinical Research Unit, Clinica Luganese Moncucco, Lugano, Switzerland. [4]Department of Biomedical Sciences, Humanitas University, Milan, Italy. [5]Department of Internal Medicine, IRCCS Humanitas Research Hospital, Milan, Italy. [6]Department of Emergency, IRCCS Humanitas Research Hospital, Milan, Italy. [7]Emergency Medicine Residency School, Department of Biomedical Sciences,  Humanitas University, Milan, Italy. [8]Department of Immunology, University Hospital Zurich, University of Zurich, Zurich, Switzerland. [9]Lyme and Tick-Borne Diseases Research Center at Columbia University Irving Medical Center, New York, NY, USA. [10]Synlab Suisse, Bioggio, Switzerland. [11]Regional Hospital Lugano, Ente Ospedaliero Cantonale, Lugano, Switzerland. [12]Università della Svizzera italiana, Lugano, Switzerland. [13]Department of Infectious Diseases, Inselspital, Bern University Hospital, University of Bern, Bern, Switzerland. [14]Rheumatology and Translational Immunology Research Laboratories (LaRIT), Division of Rheumatology, IRCCS Policlinico San Matteo Foundation, University of Pavia, Pavia, Italy. [15]Department of Rheumatology, University Hospital Zurich, University of Zurich, Zurich, Switzerland. [16]Department of Biomolecular Sciences, Biostatistics Unit, University of Urbino, Urbino, Italy. [17]Division of Infection & Immunity, Cardiff University School of Medicine, Cardiff, UK. [18]IRCCS Humanitas Research Hospital, Milan, Italy. [19]Lyme Research Program at the New York State Psychiatric Institute, New York, NY, USA. [20]Faculty of Medicine and Faculty of Science, University of Zurich, Zurich, Switzerland. [21]The William Harvey Research Institute, Queen Mary University of London, London, UK. [22]Internal Medicine and Infectious Diseases, Clinica Luganese Moncucco, Lugano, Switzerland. [23]These authors contributed equally: Jonathan Muri, Valentina Cecchinato. [24]These authors jointly supervised this work: Davide F. Robbiani and Mariagrazia Uguccioni. ✉e-mail: andrea.cavalli@irb.usi.ch; mariagrazia.uguccioni@irb.usi.ch; drobbiani@irb.usi.ch

## Methods

### Material availability

Material used in the present study is available upon request from the lead contact and may require a Material Transfer Agreement (MTA). A key resources table is provided as Supplementary Table 8.

### Study participants and ethical approvals

The Lugano COVID-19 cohort included 71 participants, diagnosed with COVID-19 at the Clinica Luganese Moncucco (CLM, Switzerland) between 8 March 2020 and 22 November 2020, who were enrolled in the study and divided into two groups, according to the severity of the acute disease. The hospitalized group included 50 participants; the outpatient group included 21 close contacts of the hospitalized group, who received only at-home care. Inclusion criteria for the hospitalized group were a SARS-CoV-2-positive nasopharyngeal swab test by quantitative PCR with reverse transcription (RT–qPCR) and age ≥ 18 yr. Inclusion criteria for the outpatient group were being a symptomatic close contact (living in the same household) of an individual enrolled in the hospitalized group and age ≥ 18 yr. Serologic tests confirmed COVID-19 positivity for all the participants (Fig. 1a and Extended Data Fig. 3a). At the 12-month visit, participants were asked to indicate the presence or absence of persisting symptoms related to COVID-19 according to a questionnaire (Supplementary Table 5). Patients who reported at least one symptom at month 12 were included in the long COVID group. The study was performed in compliance with all relevant ethical regulations and the study protocols were approved by the Ethical Committee of the Canton Ticino (ECCT): CE-3428 and CE-3960.

The Milan COVID-19 cohort included 44 participants, diagnosed with COVID-19 and hospitalized at the Humanitas Research Hospital (Milan, Italy) between 10 March 2020 and 29 March 2021, who were enrolled in the study. Inclusion criteria were a SARS-CoV-2-positive nasopharyngeal swab test by RT–qPCR and age ≥ 18 yr. Serologic tests confirmed COVID-19 positivity for the participants who were not tested by RT–qPCR. Individuals were stratified as mild or severe depending on duration of hospitalization (mild: ≤5 d; severe: ≥7 d). The study was performed in compliance with all relevant ethical regulations and the study protocols were approved by the Ethical Committee of Humanitas Research Hospital (authorization no. 738/20 and no. 956/20).

The Zurich COVID-19 cohort[7] included 104 participants, diagnosed with COVID-19 at the University Hospital Zurich, the City Hospital Triemli Zurich, the Limmattal Hospital or the Uster Hospital between April 2020 and April 2021, who were included in the study and divided into two groups, according to the severity of the acute disease. The hospitalized group included 38 participants, whereas the outpatient group included 66 individuals, who received only at-home care. Inclusion criteria for the participants were a SARS-CoV-2-positive nasopharyngeal swab test by RT–qPCR and age ≥ 18 yr. At the 13-month visit, blood was collected and participants were asked by trained study physicians to indicate the presence or absence of persisting symptoms related to COVID-19. Patients who reported at least one symptom at month 13 were included in the long COVID group. The study was performed in compliance with all relevant ethical regulations and the study protocols were approved by the Cantonal Ethics Committee of Zurich (Business Administration System for Ethics Committees (BASEC) no. 2016-01440).

The healthy control cohort included 15 adult participants (≥18 yr) with self-reported absence of previous SARS-CoV-2 infection or vaccination, enrolled between November 2020 and June 2021. An additional eight pre-pandemic samples were obtained from blood bank donors (ECCT: CE-3428). Serologic tests confirmed COVID-19 negativity for all healthy controls (Fig. 1a and Extended Data Fig. 4a). These samples were used as controls for the COVID-19 convalescents (Lugano, Milano and Zurich) and HIV-1, AS, RA and SjS cohorts.

The vaccination cohort included 16 adult participants (≥18 yr) with self-reported absence of previous SARS-CoV-2 infection (confirmed by negative serologic test; Extended Data Fig. 4f) and who received two doses of mRNA-based COVID-19 vaccine[37,38], enrolled on the day of first vaccine dose or earlier, between November 2020 and June 2021 (ECCT: CE-3428).

The HIV-1 and autoimmune diseases cohorts included pre-pandemic plasma samples obtained from the following participants: 24 HIV-1 positive (ECCT: CE-813)[39], and 13 each with AS, RA (ECCT: CE-3065, and Ethical Committee of the Canton Zurich EK-515) or SjS (Istituto di ricovero e cura a carattere scientifico (IRCCS) Policlinico San Matteo Foundation Ethics Committee no. 20070001302).

The Lyme disease cohort included plasma samples of 27 individuals with erythema migrans (Lyme disease) and 30 healthy controls obtained at The Valley Hospital (Ridgewood, NJ, USA) and the Lyme and Tick-Borne Diseases Research Center at Columbia University Irving Medical Center (New York, NY, USA) between 2015 and 2019. All were 18–89 yr of age and all denied being immunocompromised. Lyme disease cohort: individuals had new or recent onset erythema migrans, exposure to a Lyme endemic area in the previous 30 d and received no more than 3 weeks of antibiotic treatment. Healthy control cohort: individuals reported being medically healthy, had an unremarkable physical exam and blood tests, had no signs or symptoms of infection or illness, denied having had a diagnosis and/or treatment for Lyme and/or another tick-borne disease within the past 5 yr and denied having a tick bite in the previous 6 months. The Lyme cohort samples were collected at the time of the erythema migrans and 6 months later on average. The study was performed in compliance with all relevant ethical regulations and the study protocol was approved by the New York State Psychiatric Institute Institutional Review Board (no. 6805).

Written, informed consent was obtained from all participants, and all samples were coded. No compensation was provided to the study participants. Demographic and clinical features of the cohorts are reported in Supplementary Table 2.

### Blood collection, processing and storage

Blood was collected by venipuncture at approximately 6-month intervals and the PBMCs were isolated using Histopaque density centrifugation (Lugano and healthy control cohorts). Total PBMCs were aliquoted and frozen in liquid nitrogen in the presence of FCS and dimethylsulfoxide. Plasma was aliquoted and stored at −20 °C or less. Before use, plasma aliquots were heat-inactivated (56 °C for 1 h) and then stored at 4 °C. For chemotaxis assays, CD14+ monocytes and CD19+ B cells were enriched from fresh PBMCs derived from blood donors (Swiss Red Cross Laboratory; ECCT: CE-3428) through positive immunoselection (130-050-201 and 130-050-301, respectively, Miltenyi Biotec) according to the manufacturer's instructions. After isolation, CD19+ B cells were rested overnight in RPMI-1640 medium supplemented with 10% (v/v) FBS, 1% (v/v) nonessential amino acids, 1 mM sodium pyruvate, 2 mM GlutaMAX, 50 μM β-mercaptoethanol and 50 U ml$^{-1}$ penicillin/streptomycin (all from Gibco) before being used in chemotactic assays. For the other cohorts, see refs. [7,40].

### Reagents

**Peptides.** Synthetic peptides containing the N-loop or the C-terminal sequence of human chemokines were designed and obtained (>75% purity) from GenScript. All peptides are biotinylated (biotin-Ahx) at the N terminus and amidated at the C terminus. In addition, the first 2–4 amino acids of each peptide (GS, GGS, GGGS or GGK depending on the length of the N-loop/C terminus of the chemokine) consist of a linker between the biotin and the chemokine sequence. Peptides are generally 25 amino acids long, or 22–25 amino acids when synthesis was problematic. The sequence of the IFN-α2 peptide (7–28) was based on a previously described immunoreactive epitope in patients with myasthenia gravis[41], and that of the SARS-CoV-2 nucleocapsid protein (N) peptide (157–178) was described in ref. [42]. An irrelevant peptide was used as negative control. The amino acid sequences of all peptides in this study are listed in Supplementary Table 3.

**Proteins.** CCL7, CCL20, CXCL8 and CXCL13 were synthesized using tBoc solid-phase chemistry[43]. CCL8 and CXCL16 were obtained from Peprotech (catalog no. 300-15 and catalog no. 300-55, respectively) or produced and purified in-house. Briefly, recombinant chemokines were expressed in *E. coli*, purified from inclusion bodies by immobilized-metal affinity chromatography and folded under $N_2$ protection in an arginine-containing buffer (80 mM Tris-Cl (pH 8.5), 100 mM NaCl, 0.8 M arginine, 2 mM EDTA, 1 mM cysteine, 0.2 mM cystine) as previously described[44]. After recovery and concentration, the purification tag was cleaved with enterokinase, and the processed chemokine was purified by $C_{18}$ reverse-phase chromatography. The SARS-CoV-2 RBD was produced and purified as described[45].

## Chemotaxis

The migration of primary human monocytes and B cells isolated from buffy coats or of murine preB 300.19 cells stably expressing the human chemokine receptors CCR2 (ref. [46]), CCR6, CXCR1 (ref. [47]) and CXCR6 (ref. [48]) was assayed using 48-well Boyden chambers (Neuro Probe) with polyvinylpyrrolidone-free polycarbonate membranes with pore size of 3 μm for primary human B cells and 5 μm for the other cell types, as previously described[49]. Briefly, $10^5$ primary human B cells or $5 \times 10^4$ primary human monocytes and murine preB 300.19 cells were diluted in RPMI-1640 supplemented with 20 mM HEPES, pH 7.4, and 1% pasteurized plasma protein solution (5% PPL SRK; Swiss Red Cross Laboratory). Cells were then added to the upper wells and the chemokine (with or without antibodies) to the bottom wells. After 120 min of incubation for primary human B cells and 90 min for the other cell types, the membrane was removed, washed on the upper side with PBS, fixed and stained with DiffQuik. All assays were done in triplicate, and for each well the migrated cells were counted at 100-fold magnification in five randomly selected high-power fields.

**Inhibition of chemotaxis by monoclonal antibodies.** Experiments were performed with monoclonal antibodies at a final concentration of 30 μg ml⁻¹ (Extended Data Fig. 8g) or 50 μg ml⁻¹ (Fig. 2e,g and Extended Data Fig. 8j). Baseline migration was determined in the absence of chemoattractant (buffer control).

**Inhibition of chemotaxis by plasma purified IgGs.** IgGs were purified from a subset of samples of the COVID-19 and healthy control cohorts using Protein G Sepharose 4 Fast Flow (Cytiva) according to the manufacturer's instructions (plasma/resuspended beads at a 5:4 (v/v) ratio), buffer-exchanged and concentrated in PBS by Amicon Ultra-4 centrifugal filters (30-kDa cutoff, Millipore). Chemotaxis of preB 300.19 expressing CCR2 or CXCR1 was performed at a final IgG concentration of 200 μg ml⁻¹ (IgG concentration in human serum: ~10,000 μg ml⁻¹ (ref. [29])), in the presence of the chemokine concentration resulting in peak migration when no antibodies were added (CCL7 (100 nM), CCL8 (100 nM), CXCL8 (1 nM)) (Extended Data Fig. 8k).

## ELISA

To evaluate the antibodies' binding to chemokine peptides, 96-well plates (ThermoFisher, 442404) or 384-well plates (ThermoFisher, 464718) were coated with 50 μl (or 10 μl for 384-well plates) per well of a 2 μg ml⁻¹ Neutravidin (Life Technologies, 31000) solution in PBS, overnight at room temperature. Plates were washed four times with washing buffer (PBS + 0.05% Tween-20 (Sigma-Aldrich)) and incubated with 50 μl (or 10 μl for 384-well plates) per well of a 50 nM biotinylated peptide solution in PBS for 1 h at room temperature. After washing four times with washing buffer, plates were incubated with 200 μl (or 50 μl for 384-well plates) per well of blocking buffer (PBS + 2% BSA + 0.05% Tween-20) for 2 h at room temperature. Plates were then washed four times with washing buffer, and serial dilutions of monoclonal antibodies or plasma were added in PBS + 0.05% Tween-20 and incubated for 1 h at room temperature. To screen for the presence of chemokine

IgGs, plasma samples were assayed (unless otherwise stated) at 1:50 starting dilution followed by three fourfold serial dilutions (1:200, 1:800, 1:3,200). Monoclonal antibodies were tested at 5 μg ml⁻¹ starting concentration followed by 11 threefold serial dilutions. Plates were subsequently washed four times with washing buffer and incubated with an antibody against human IgG secondary antibody conjugated to horseradish peroxidase (HRP) (GE Healthcare, NA933) at a 1:5,000 dilution in PBS + 0.05% Tween-20. Finally, after washing four times with washing buffer, plates were developed by the addition of 50 μl (or 10 μl for 384-well plates) per well of the HRP substrate tetramethylbenzidine (TMB) (ThermoFisher, 34021) for 10 min. The developing reaction was stopped with 50 μl (or 10 μl for 384-well plates) per well of a 1 M $H_2SO_4$ solution, and absorbance was measured at 450 nm with an ELISA microplate reader (BioTek) with Gen5 3.12 software. A positive control (broadly reactive plasma from donor CLM70) and negative control (uninfected participant) samples were included in each experiment. Since the basal average optical density likely also depends on intrinsic features of each peptide that is used to coat the ELISA plate, the presented values should be interpreted as relative rather than absolute. The area under the curve (AUC) was obtained from two independent experiments and plotted with GraphPad Prism v.9.0.2. The main findings were further confirmed by assaying subsets of samples belonging to the different groups, side-by-side on the same plates.

**Lyme disease cohort.** Plasma was assayed at a 1:100 starting dilution, followed by two additional fourfold dilutions (1:400 and 1:1,600) (Extended Data Fig. 10a).

**Reactivity at month 6 versus 12.** Experiments were performed with plasma samples from different time points side-by-side on the same plate. In Extended Data Fig. 4b, plasma was assayed at a 1:50 starting dilution, followed by four additional fivefold dilutions. RBD IgG antibody levels were measured in COVID-19 convalescents who had not received a COVID-19 mRNA vaccine between first and second visits (no vaccination) or in individuals with at least one dose of vaccine at least 10 d before blood sampling at the second visit (Fig. 1d, Extended Data Fig. 4b,c and Supplementary Table 2).

**Kinetics of signature chemokine IgG antibodies.** Experiments were performed with plasma samples from different time points assayed at 1:50 dilution side-by-side on the same plate, and the average optical density at 450 nm obtained from two independent experiments was plotted with GraphPad Prism v.9.0.2 (Extended Data Fig. 4e).

**IgG antibodies binding to SARS-CoV-2 RBD.** Experiments were performed with 96-well plates coated with 50 μl per well of a 5 μg ml⁻¹ protein solution in PBS overnight at room temperature, and subsequently blocked and treated as described above. In this case, plasma samples were assayed at a 1:50 starting dilution either followed by seven additional threefold serial dilutions (Figs. 1a and 2a and Extended Data Figs. 3a,c,d and 6a) or followed by three additional fivefold serial dilutions (Extended Data Fig. 4b,f).

## Chemokine quantification in plasma

Plasma levels of 14 chemokines were measured using the Luminex Discovery Assay–Human Premixed Multi-Analyte Kit (R&D Systems, LXSAHM-14) following the manufacturer's instructions. Chemokines included in the panel were: CCL2, CCL3, CCL4, CCL19, CCL21, CCL22, CCL25, CXCL2, CXCL5, CXCL8, CXCL9, CXCL10, CXCL13 and CXCL16. Each sample was measured in duplicate using a Luminex FLEXMAP 3D system.

## Single-cell sorting by flow cytometry

B cells were enriched from PBMCs of healthy controls or of COVID-19 convalescents 6 months after COVID-19 (participant CLM9 for CCL8

antibodies; CLM64 for CCL20 antibodies; CLM5, CLM7 and CLM33 for CXCL13 antibodies; and CLM8 and CLM30 for CXCL16 antibodies), using the pan-B-cell isolation kit according to manufacturer's instructions (Miltenyi Biotec, 130-101-638). The enriched B cells were subsequently stained in FACS buffer (PBS + 2% FCS + 1 mM EDTA) with the following antibodies/reagents (all 1:200 diluted) for 30 min on ice: antibodies against CD20-PE-Cy7 (BD Biosciences, 335828), against CD14-APC-eFluor 780 (ThermoFisher, 47-0149-42), against CD16-APC-eFluor 780 (ThermoFisher, 47-0168-41), against CD3-APC-eFluor 780 (ThermoFisher, 47-0037-41), against CD8-APC-eFluor 780 (Invitrogen, 47-0086-42); Zombie NIR (BioLegend, 423105); as well as fluorophore-labeled ovalbumin (Ova) and N-loop peptides. Live single Zombie-NIR⁻CD14⁻CD16⁻CD3⁻CD8⁻CD20⁺Ova⁻N-loop-PE⁺N-loop-AF647⁺ B cells were single-cell sorted into 96-well plates containing 4 µl of lysis buffer (0.5 × PBS, 10 mM DTT, 3,000 U ml⁻¹ RNasin Ribonuclease Inhibitors (Promega, N2615)) per well using a FACS Aria III, and the analysis was performed with FlowJo software. The CCL20 antibody sequences were obtained by sorting with a pool of 12 peptides; for all the others, a single peptide was used. The sorted cells were frozen on dry ice and stored at −80 °C.

### Antibody sequencing, cloning, production and purification
Antibody genes were sequenced, cloned and expressed as previously reported[50–52]. Briefly, reverse transcription of RNA from FACS-sorted single cells was performed to obtain complementary DNA, which was then used for amplification of the immunoglobulin IGH, IGK and IGL genes by nested PCR. Amplicons from this first PCR reaction served as templates for sequence- and ligation-independent cloning into human IgG1 antibody expression vectors. Monoclonal antibodies were produced by transiently transfecting Expi293F cells cultured in Freestyle-293 Expression Medium (ThermoFisher) with equal amounts of immunoglobulin heavy and light chain expression vectors using polyethyleneimine Max (PEI-MAX, Polysciences) as a transfection reagent. After 6–7 d of culture, cell supernatants were filtered through 0.22-µm Millex-GP filters (Merck Millipore), and antibodies were purified using Protein G Sepharose 4 Fast Flow (Cytiva) according to the manufacturer's instructions and buffer-exchanged and concentrated in PBS by Amicon Ultra-4 centrifugal filters (30-kDa cutoff, Millipore). Where indicated, the monoclonal antibody against Zika virus Z02150 was used as an isotype control.

### Computational analysis of antibody sequences
Antibody sequences were analyzed using a collection of Perl and R scripts provided by IgPipeline and publicly available on GitHub (https://github.com/stratust/igpipeline)[27]. In brief, sequences were annotated using IgBlast[53] v.1.14.0 with IMGT domain delineation system and the Change-O toolkit v.0.4.5 (ref. [54]). CDR3 sequences were determined by aligning the IGHV and IGLV nucleotide sequences against their closest germlines using the blastn function of IgBlast.

### SARS-CoV-2 pseudotyped reporter virus and neutralization assay
To generate (HIV-1/NanoLuc2AEGFP)-SARS-CoV-2 particles, HEK293T cells were co-transfected with the three plasmids pHIV$_{NL}$Gag-Pol, pCCNanoLuc2AEGFP and SARS-CoV-2 S as described elsewhere[27,55]. Supernatants containing virions were collected 48 h after transfection, and virion infectivity was determined by titration on 293T$_{ACE2}$ cells. The plasma neutralizing activity was measured as previously reported[27,55]. Briefly, threefold serially diluted plasma samples (from 1:50 to 1:328,050) were incubated with SARS-CoV-2 pseudotyped virus for 1 h at 37 °C, and the virus–plasma mixture was subsequently incubated with 293T$_{ACE2}$ cells for 48 h. Cells were then washed with PBS and lysed with Luciferase Cell Culture Lysis 5X reagent (Promega). Nanoluc Luciferase activity in cell lysates was measured using the Nano-Glo Luciferase Assay System (Promega) with Modulus II Microplate Reader

user interface (TURNER BioSystems). The obtained relative luminescence units were normalized to those derived from cells infected with SARS-CoV-2 pseudotyped virus in the absence of plasma. The NT$_{50}$ values were determined using four-parameter nonlinear regression with bottom and top constrains equal to 0 and 1, respectively (GraphPad Prism v.9.0.2). The dotted line (NT$_{50}$ = 5) in the plots represents the lower limit of detection of the assay.

### Model interaction between chemokine and chemokine receptor
The illustrative model in Extended Data Fig. 1a was generated from the structure of inactive CCR2 (PDB code: 5T1A)[56], together with the electron microscopy structures of CCR5 and CCR6 (PDB codes: 6MEO and 6WWZ, respectively)[57,58], by using the SWISS-MODEL[59] server and the molecular graphics program PyMOL 2.5.0 for modeling the N and C termini of the receptor. The crystal structure of CCL8 (MCP-2) (PDB code: 1ESR)[60] and the electron microscopy structure of CCR6 (ref. [58]) were used to model the complex. The intracellular residues were removed for clarity.

### Statistical analysis
**Sample size definition.** No statistical methods were used to pre-determine sample sizes but our sample sizes are similar to those reported in previous publications[7].

**Tests for statistical significance.** Upon testing of parametric assumptions (Kolmogorov–Smirnov test for normality and Hartley's $F_{max}$ test for homoskedasticity), statistical significance between two groups was determined using parametric paired two-tailed Student's $t$-test, or nonparametric two-tailed Mann–Whitney $U$-tests (unpaired samples), or Wilcoxon signed-rank test (paired samples). Statistical significance between more than two groups was evaluated using Kruskal–Wallis test (followed by Dunn multiple comparisons), one-way analysis of variance (ANOVA) (followed by Tukey multiple comparisons) or two-way repeated measures ANOVA (followed by Šídák multiple comparisons), as described in the figure legends. Statistical significance of the signature chemokines (CCL19, CCL22, CXCL17, CXCL8, CCL25, CXCL5, CCL21, CXCL13 and CXCL16) was also confirmed when applying the Bonferroni criterion to guarantee a familywise level of significance equal to 0.05. Statistical significance from a 2 × 2 contingency table was determined with Fisher's exact test. Correlations were assessed using Pearson correlation analysis. A $P$ value of less than 0.05 was considered statistically significant. Data and statistical analyses were performed with GraphPad Prism v.9.0.2. Data collection and analysis were not performed blind to the conditions of the experiments.

**$t$-SNE.** $t$-SNE analysis was performed using the Rtsne R package v.0.15 (https://CRAN.R-project.org/package=Rtsne) using the AUC values for all chemokines. The theta parameter for the accuracy of the mapping was set to zero in all cases for exact $t$-SNE.

**Clustering.** Hierarchical clustering was created using the hclust R function v.4.1.1. Clustering analysis was performed using the correlation as distance and Ward's method for agglomeration. Heatmaps were created with either GraphPad Prism v.9.0.2 (Fig. 1a and Extended Data Fig. 2d) or the *Pretty Heatmaps* (*pheatmap*) R package v.1.0.12 (Extended Data Fig. 10b). In Extended Data Fig. 10b, each column containing a distinct chemokine was scaled with the scaling function provided by R, which sets the mean and the standard deviation to 0 and 1, respectively.

**Logistic regression and additional analyses.** Logistic regression was performed using the GLM (Generalized Linear Models) function provided by the R package v.4.1.1. To identify which variables to include in the analysis, AUCs were ranked according to the $P$ value obtained with a Mann–Whitney–Wilcoxon nonparametric test on the Lugano cohort.

# Letter

The first $N$ variables minimizing the Akaike information criterion were then used in the fitting. Furthermore, the same set of variables was used to perform the fitting with the Milan and Zurich cohorts. In each plot, values from 0 to 0.5 and from 0.5 to 1 on the $y$ axis represent the assignment of individuals to the A and B groups (of a Prediction A versus B; see gray backgrounds), respectively. On the $x$ axis, samples are divided into the two groups and subsequently ordered according to sample identity as shown in Supplementary Table 2. Dots in the gray area represent individuals that are assigned to the correct group. We additionally performed chi-tests considering covariates that are known to influence COVID-19 severity (demographics (sex and age) and comorbidities (diabetes and cardiovascular diseases)) and found that none of them was significantly different between groups. Race/ethnicity was not analyzed because the cohort is nearly 100% White; similarly, immune deficiency was rare and was not considered. Logistic regression analysis using the combination of these covariates (age, sex, diabetes and cardiovascular diseases) allowed proper assignment with accuracies of 74.6% (COVID-19 severity; outpatient versus hospitalized) and 68.3% (long COVID; no long COVID versus long COVID). Notably, the accuracy using chemokine antibody values is even better (77.5% (COVID-19 severity) and 77.8% (long COVID)). These analyses are shown in Supplementary Table 9.

## Reporting summary

Further information on research design is available in the Nature Portfolio Reporting Summary linked to this article.

## Data availability

All data analyzed during the present study are included in this article and its supporting information files. Source data are provided with this paper. PDB accession codes are 5T1A, 6MEO, 6WWZ and 1ESR.

## Code availability

Computer code for antibody sequence, logistic regression, clustering and $t$-SNE analyses has been deposited at GitHub (https://github.com/cavallilab/chemopept).

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

## Acknowledgements

We thank all study participants and their families; the medical personnel of the Clinica Luganese Moncucco; C. Sotsky, N. Rioux, S. Doshi and T. Hijazi (Lyme disease cohort); M. Sironi, R. Leone, M. Rimoldi and the Humanitas COVID-19 Task Force (Milan cohort); S. Hasler, S. Adamo, M. Raeber, J. Nilsson, E. Bächli, A. Rudiger and L. Huber (Zurich cohort); and S. Jovic (Institute for Research in Biomedicine) for their support and technical assistance. We also acknowledge T. Oliveira (Rockefeller University) for sharing the script for the clustering analysis, and T. Hatziioannou and P. Bieniasz (Rockefeller University) for sharing plasmids and protocols for SARS-CoV-2 pseudovirus. A. Cavalli thanks F. Gruner for the generous support. This study was also in part financed within the framework of the Swiss HIV Cohort Study using data gathered by the Five Swiss University Hospitals, two Cantonal Hospitals, 15 affiliated hospitals and 36 private physicians (listed in http://www.shcs.ch/180-health-care-providers). Members of the Swiss HIV Cohort Study are listed in https://shcs.ch/184-for-shcs-publications. We are grateful to M. Baggiolini for mentoring, support and fruitful discussions. This work was supported by the Swiss Vaccine Research Institute (to D.F.R.), the National Institutes of Health (grants no. U01 AI151698—UWARN

United World Antiviral Research Network, no. P01 AI138938 and no. U19 AI111825 to D.F.R.); the European Union's Horizon 2020 research and innovation programme (grant no. 101003650 to D.F.R. and L.V.); BRIDGE no. 40B2-0_203488 (to A. Cavalli and D.F.R.); the Fidinam Foundation (to M. Uguccioni); the Rocca Foundation (to M. Uguccioni and A. Mantovani); the Ceschina Foundation (to M. Uguccioni); the Swiss HIV Cohort Study (grant no. 719), the Swiss National Science Foundation (grant no. 201369) and SHCS Research Foundation (to M. Uguccioni, E.B. and A.R.); the Italian Ministry of Health for COVID-19 (grant no. COVID-2020-12371640 to A. Mantovani); Dolce & Gabbana fashion house (to A. Mantovani); the Swiss National Science Foundation (grant no. NRP 78 Implementation Programme to C.C. and O.B; grants no. 4078P0-198431, no. 310030-200669 and no. 310030-212240 to O.B.); Clinical Research Priority Program CYTIMM-Z of the University of Zurich (UZH; to O.B.); the Pandemic Fund of UZH (to O.B.); the Innovation grant of USZ (to O.B.); the Digitalization Initiative of the Zurich Higher Education Institutions Rapid-Action (Call no. 2021.1_RAC_ID_34 to C.C.); the Swiss Academy of Medical Sciences (SAMW; fellowships no. 323530-191220 to C.C. and no. 323530-191230 to Y.Z.); and the Lyme and Tick-Borne Diseases Research Center at Columbia University Irving Medical Center, established by the Global Lyme Alliance, Inc. and the Lyme Disease Association, Inc. (to B.A.F.).

## Author contributions

J. Muri, V. Cecchinato, A. Cavalli, M. Uguccioni and D.F.R. conceived the study. M.M., S.M., J.S. and A. Cavalli developed softwares. J. Muri, V. Cecchinato and A. Cavalli validated the data. J. Muri, V. Cecchinato, A. Cavalli, S.M. and M.B.L.R. performed formal analyses. J. Muri, V. Cecchinato, A.A.S., E.G., J. Moritz, T.G. and F. Bianchini performed the experiments. J. Muri, T.G., P.P., F. Bianchini, V. Crivelli, L.P., C.T., G.D.-S., B.M., M.T., D.J., S.T., M.P., L.V., M.B., P.A.M., A.F.-P., C.G., M. Uhr, E.B., A.R., A. Ciurea, A. Manzo, R.F., F. Barbic, A.V., G.D.N., C.C., P.T., Y.Z., L.A.M., B.B., A. Mantovani, B.A.F. and O.B. provided resources.

J. Muri, V. Cecchinato, M. Uguccioni and D.F.R. wrote the original and revised manuscripts. D.F.R., M. Uguccioni, A. Cavalli, L.V., A.R., E.B., B.A.F., O.B., C.C. and A. Mantovani acquired funding. D.F.R. and M. Uguccioni supervised and equally contributed to this work.

## Funding

## Competing interests

The Institute for Research in Biomedicine has filed a provisional patent application in connection with this work on which J. Muri, V. Cecchinato, A. Cavalli, M. Uguccioni and D.F.R. are inventors. The remaining authors declare no competing interests.

## Additional information

**Extended data** is available for this paper at https://doi.org/10.1038/s41590-023-01445-w.

**Correspondence and requests for materials** should be addressed to Andrea Cavalli, Mariagrazia Uguccioni or Davide F. Robbiani.

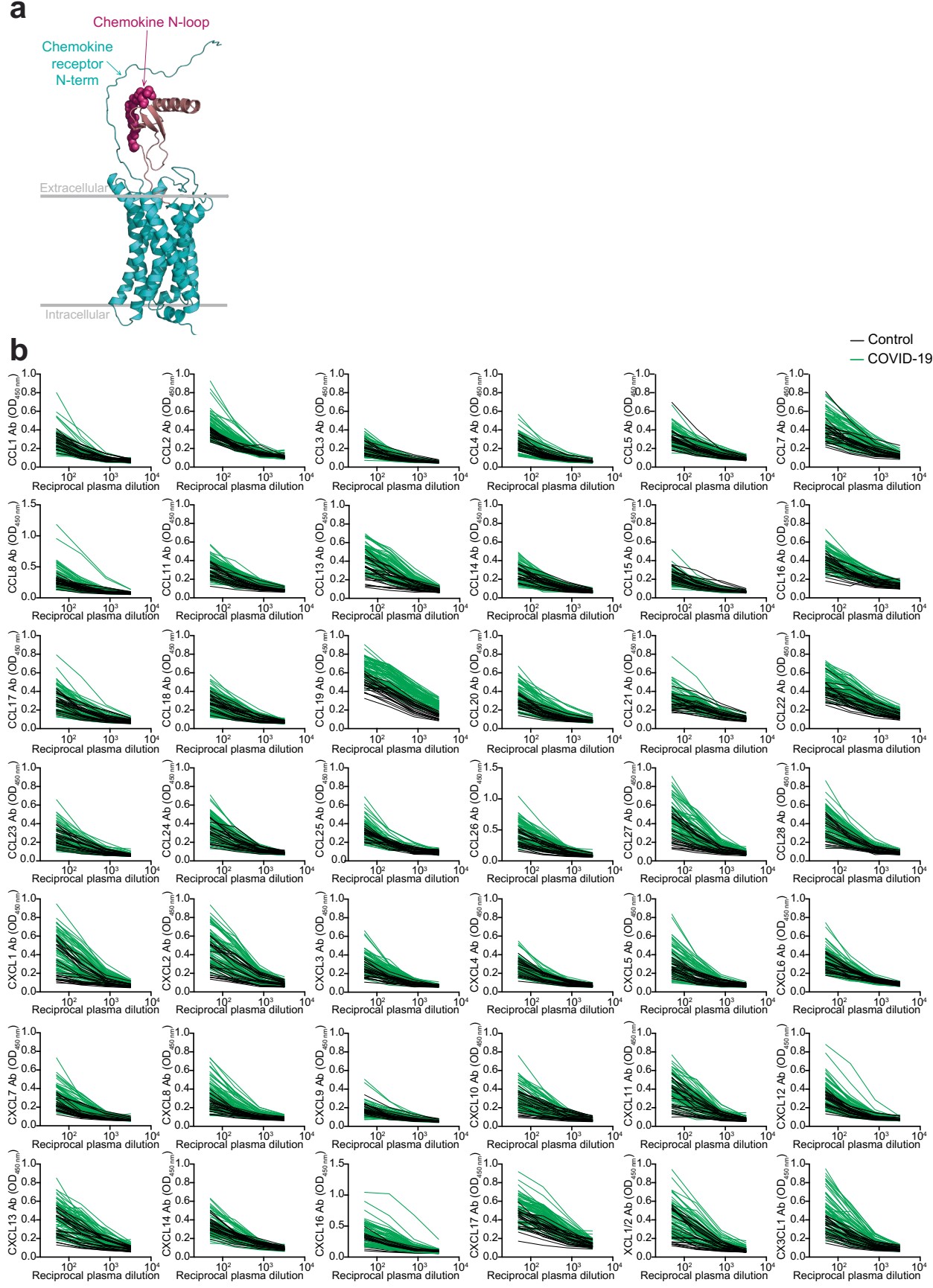

**Extended Data Fig. 1 | See next page for caption.**

**Extended Data Fig. 1 | Chemokine N-loop antibodies in COVID-19 convalescents.** (**a**) Model showing the interaction between a chemokine and its receptor. Arrows point to the area of putative interaction between the N-terminus of the receptor and the chemokine N-loop (shown by spheres). Chemokine is magenta and chemokine receptor is cyan. (**b**) ELISA curves showing the levels of chemokine N-loop antibodies in healthy controls (Control, n = 23) and COVID-19 convalescents (COVID-19, n = 71) from the Lugano cohort at month 6. Average optical density ($OD_{450}$) measurements of two independent experiments.

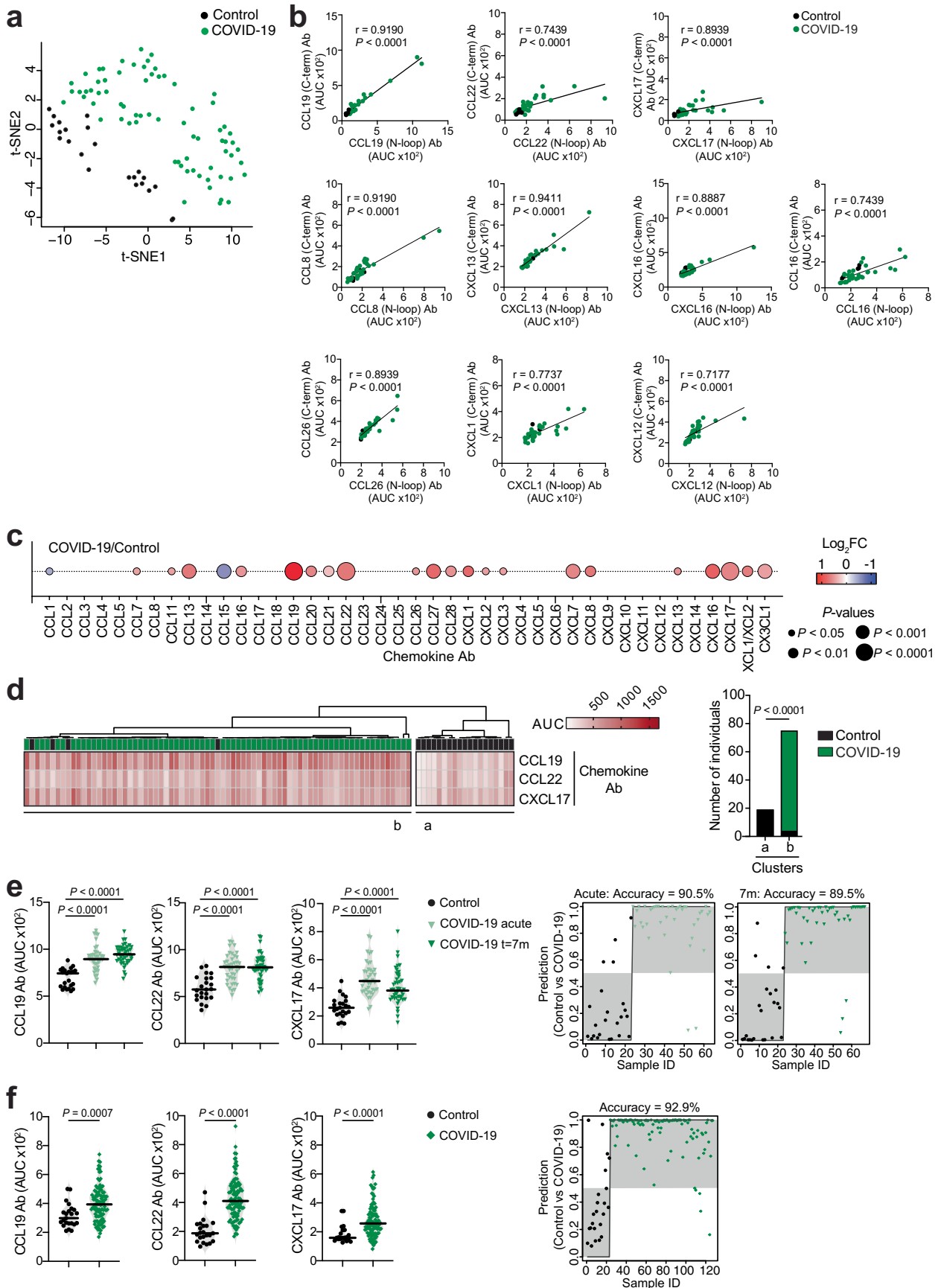

**Extended Data Fig. 2 | See next page for caption.**

**Extended Data Fig. 2 | Analyses of chemokine antibodies in COVID-19 convalescents.** (**a**) t-SNE distribution of healthy controls (Control, n = 23) and COVID-19 convalescents from the Lugano cohort at month 6 (COVID-19, n = 71), as determined with the 42 datasets combined. (**b**) Correlations of antibodies to the N-loop and C-terminal peptides of the same chemokine by two-tailed Pearson correlation analysis. ELISA was performed on a subset of samples (healthy controls, n = 5; COVID-19 convalescents from the Lugano cohort at month 6, n = 31). Average of two independent experiments. (**c**) Chemokine antibodies in COVID-19 convalescents from the Lugano cohort at month 6 (n = 71), shown as ratio over healthy controls (n = 23). Circle size indicates significance; colors show the $Log_2$ fold-change increase (red) or decrease (blue), shown as ratio over healthy controls. Two-tailed Mann–Whitney U-tests. (**d**) Unsupervised hierarchical clustering analysis with CCL19, CCL22 and CXCL17 antibodies showing the distribution of COVID-19 convalescents (month 6, Lugano cohort) and healthy controls in two separate clusters. Two-tailed Fisher's exact test. (**e**) Left, AUC of ELISA showing CCL19, CCL22 and CXCL17 antibodies in healthy controls (n = 23) and COVID-19 Milan cohort during acute disease (n = 40) and at month 7 (n = 44). Kruskal-Wallis test followed by Dunn's multiple comparison test. Right, logistic regression analysis assignment of the COVID-19 Milan cohort and healthy controls based on CCL19, CCL22 and CXCL17 antibodies during acute disease and at month 7. (**f**) Same as in (**e**) but for the Zurich cohort at month 13. Healthy controls (n = 23), and COVID-19 convalescents (n = 104). Two-tailed Mann–Whitney U-tests.

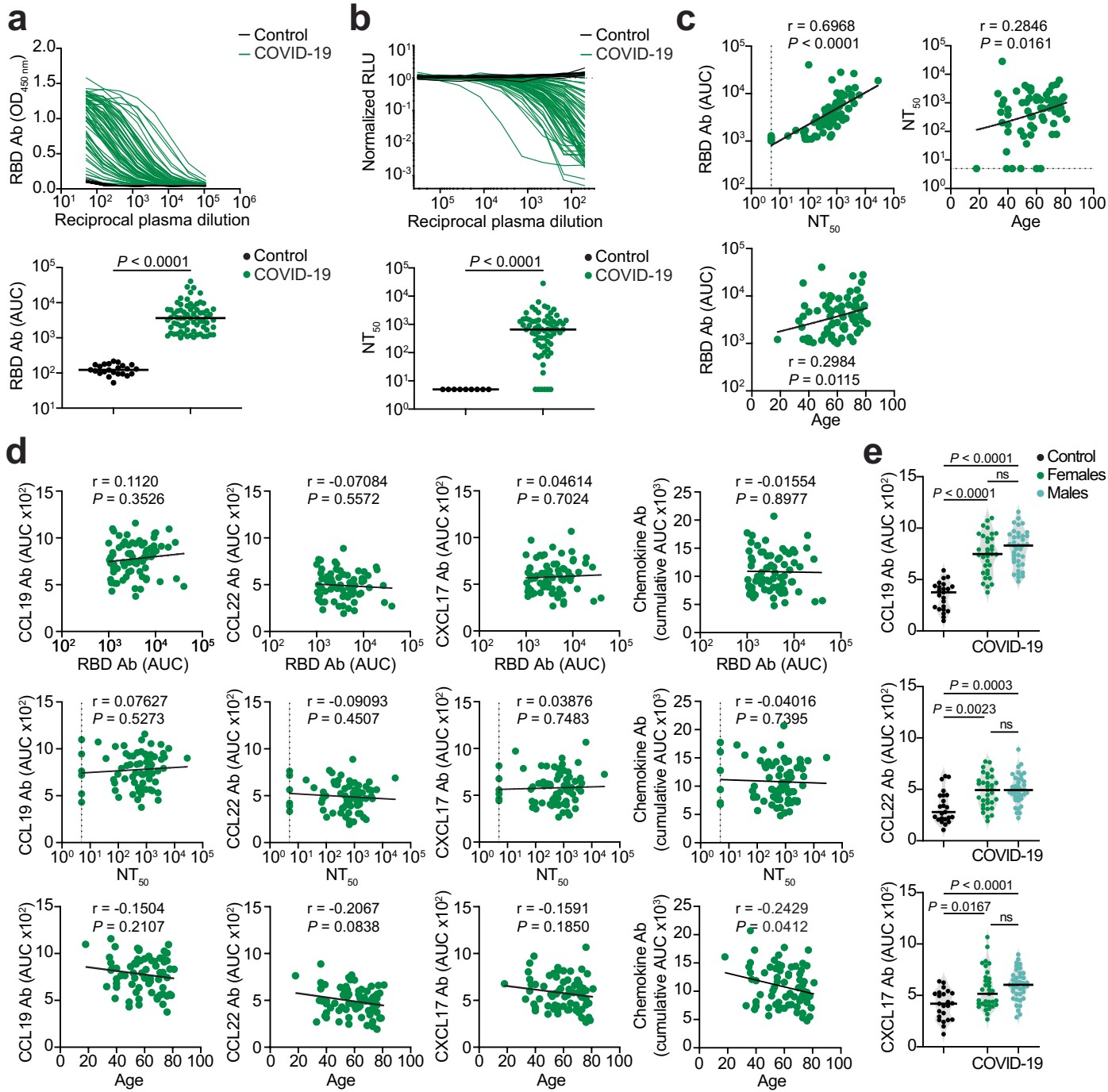

**Extended Data Fig. 3 | Correlation analyses of chemokine antibodies.**
(**a**) ELISA showing RBD IgG antibodies as $OD_{450}$ of serial plasma dilutions (top panel) and as AUC (bottom panel) in healthy controls (Control, n = 23) and COVID-19 convalescents (COVID-19, n = 71) from the Lugano cohort at month 6. Average of two independent experiments. Horizontal bars indicate median values. Two-tailed Mann–Whitney U-tests. (**b**) Neutralizing activity against SARS-CoV-2 pseudovirus shown as relative luciferase units (RLU) normalized to no plasma control (top panel) and half-maximal neutralizing titers ($NT_{50}$, bottom panel) in healthy controls (n = 9) and in COVID-19 convalescents (n = 71) from the Lugano cohort at month 6. Average of two independent experiments.

Horizontal bars indicate median values. Two-tailed Mann–Whitney U-tests.
(**c**) Two-tailed Pearson correlations of RBD IgG and $NT_{50}$ values to each other and with age. Average of two independent experiments. (**d**) Two-tailed Pearson correlations of CCL19, CCL22, CXCL17 antibodies and of the cumulative signal of the 42 chemokine antibodies with RBD IgG, $NT_{50}$ values and age. (**e**) AUC of ELISA showing CCL19, CCL22 and CXCL17 antibodies in healthy controls (n = 23) and COVID-19 convalescents (n = 71) from the Lugano cohort at month 6 grouped by gender (n = 33, females; n = 38, males). Data are shown as average of two independent experiments. Horizontal bars indicate median values. Kruskal-Wallis test followed by Dunn's multiple comparison test.

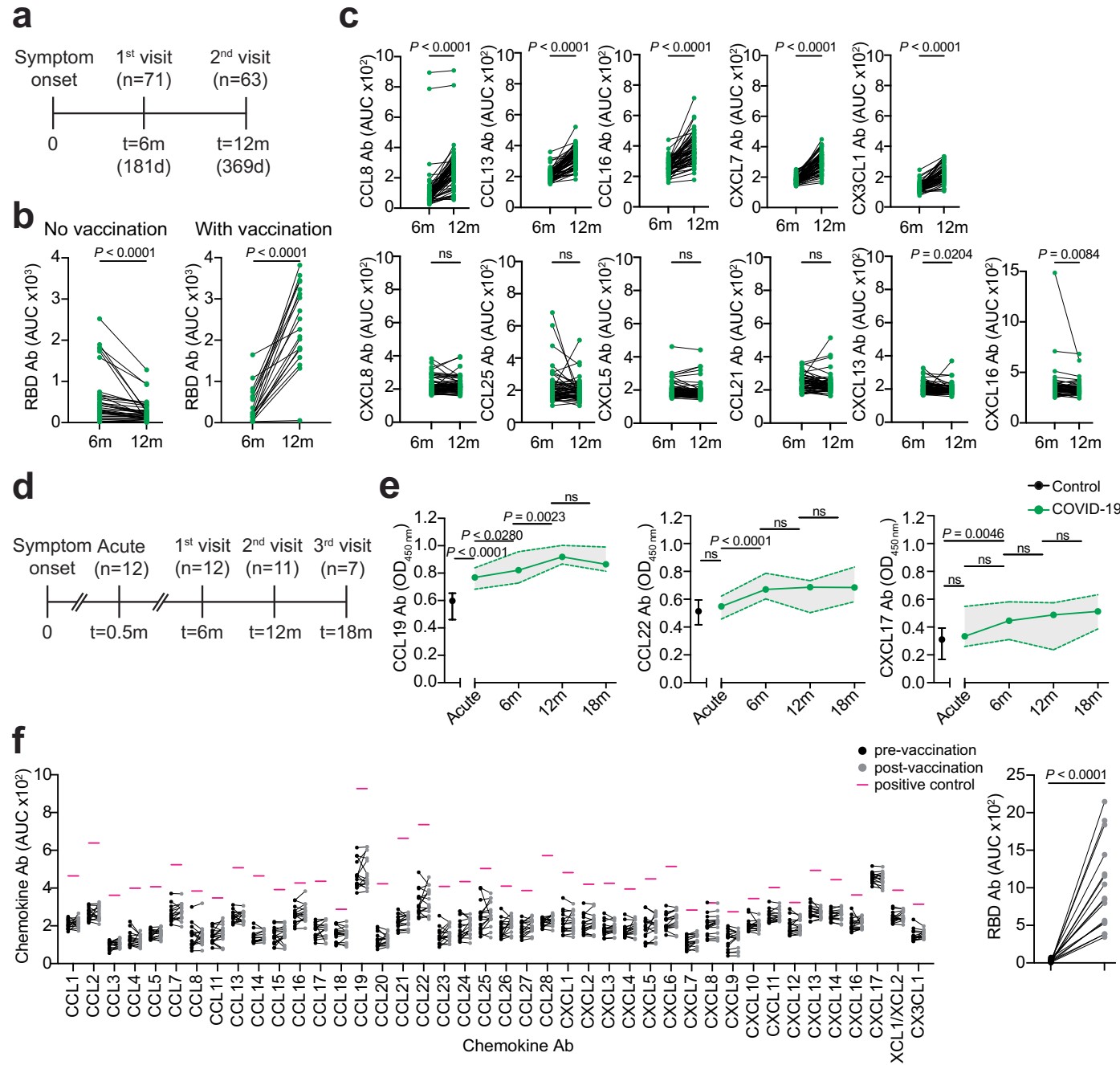

**Extended Data Fig. 4 | Chemokine antibodies in COVID-19 convalescents over time and upon COVID-19 vaccination.** (**a**) Diagram of the time points of blood collection after onset of COVID-19 symptoms in the Lugano cohort. (**b**) AUC of ELISA showing RBD IgG antibodies at month 6 and 12 in vaccinated (n = 19) and non-vaccinated (n = 40) COVID-19 convalescents from the Lugano cohort. Average from two independent experiments. Two-tailed Wilcoxon signed-rank test. (**c**) AUC of ELISA showing chemokine antibodies in COVID-19 convalescents from the Lugano cohort at month 6 and 12 (n = 63). Two independent experiments. Two-tailed Wilcoxon signed-rank test. (**d**) Diagram of the time points of blood collection after onset of COVID-19 symptoms in a subset of previously hospitalized COVID-19 convalescents from the Lugano cohort.

(**e**) ELISA showing CCL19, CCL22 and CXCL17 antibodies in healthy controls (Control, n = 10) and COVID-19 convalescents (COVID-19) from the Lugano cohort at day 15 (acute, n = 12), and at month 6 (n = 12), 12 (n = 11) and 18 (n = 7). Average OD$_{450}$ values from two independent experiments. One-way ANOVA test followed by Tukey's multiple comparison test. Data are shown as median±range. (**f**) AUC of ELISA showing chemokine antibodies in SARS-CoV-2 naïve individuals (n = 16) before and at month 4 on average after COVID-19 mRNA vaccination. Two independent experiments. Pink lines represent the signal of a positive control plasma sample with broad reactivity (CLM70). RBD IgG is shown alongside as control (right panel). Two-tailed Wilcoxon signed-rank test with false discovery rate (FDR) approach.

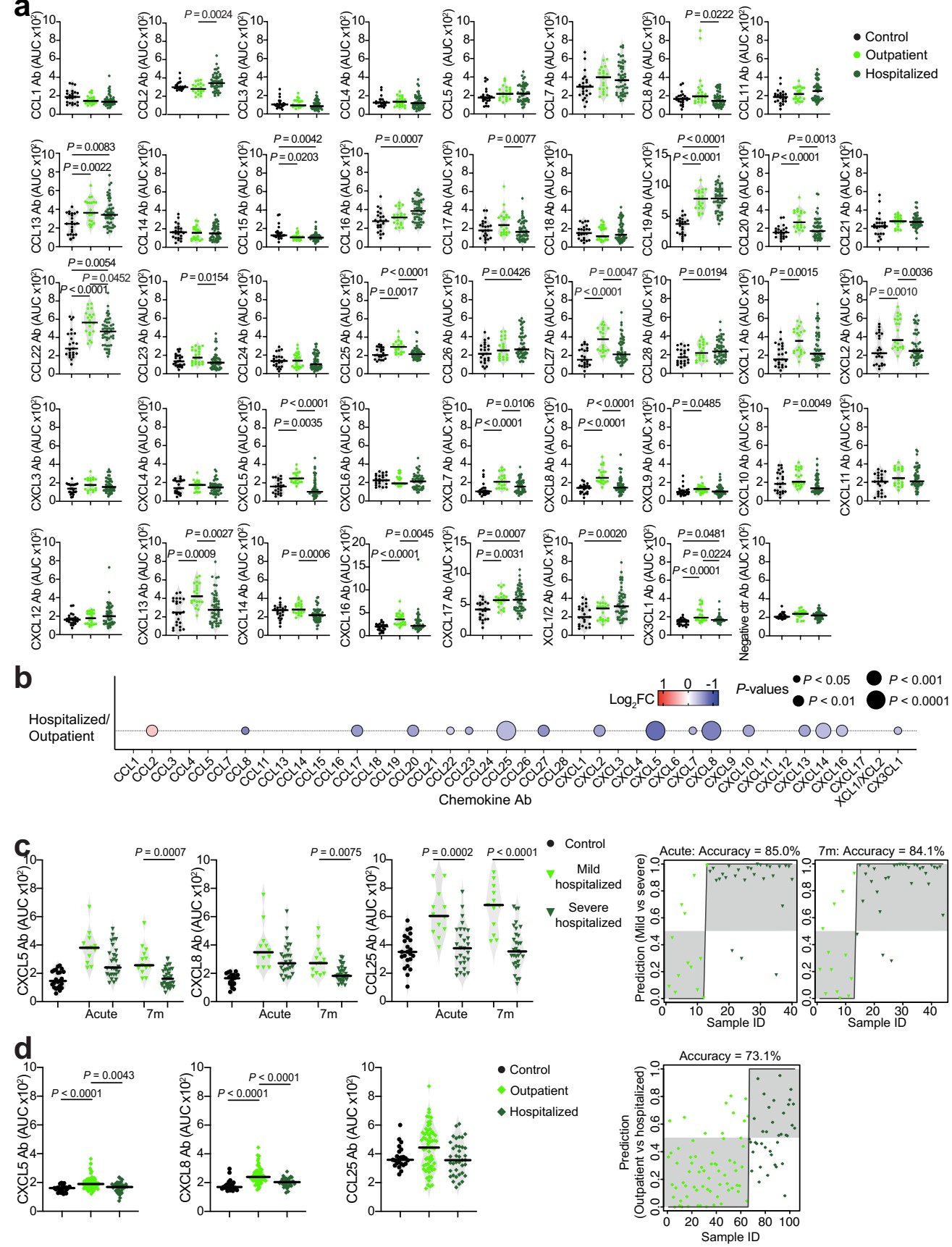

**Extended Data Fig. 5 | See next page for caption.**

**Extended Data Fig. 5 | Chemokine antibodies in previously hospitalized and outpatient COVID-19 convalescents.** (**a**) AUC of ELISA showing chemokine antibodies in healthy controls (Control, n = 23) and COVID-19 convalescents from the Lugano at month 6 that were previously hospitalized (n = 50) or outpatient (n = 21). Average of two independent experiments. Horizontal bars indicate median values. Kruskal-Wallis test followed by Dunn's multiple comparison test. (**b**) Chemokine antibodies in previously hospitalized (n = 50), shown as ratio over outpatient (n = 21) COVID-19 convalescents of the Lugano cohort at month 6. Circle size indicates significance; colors show the $\text{Log}_2$ fold-change increase (red) or decrease (blue), shown as ratio over outpatient COVID-19 convalescents. Kruskal-Wallis test followed by Dunn's multiple comparison test. (**c**) Left, AUC of ELISA showing CXCL5, CXCL8 and CCL25 antibodies in healthy controls (n = 23) and in mild (n = 13, acute; n = 27, month 7) versus severe (n = 27, acute; n = 31, month 7) hospitalized COVID-19 from the Milan cohort during acute disease and at month 7. Kruskal-Wallis test followed by Dunn's multiple comparison test. Right, logistic regression analysis assignment of mild and severe hospitalized COVID-19 convalescents from the Milan cohort based on CXCL5, CXCL8 and CCL25 antibodies during acute disease and at month 7. (**d**) Same as in (**c**) but for the Zurich cohort at month 13. Healthy controls (n = 23) and COVID-19 convalescents that were previously hospitalized (n = 38) or outpatient (n = 66). Kruskal-Wallis test followed by Dunn's multiple comparison test.

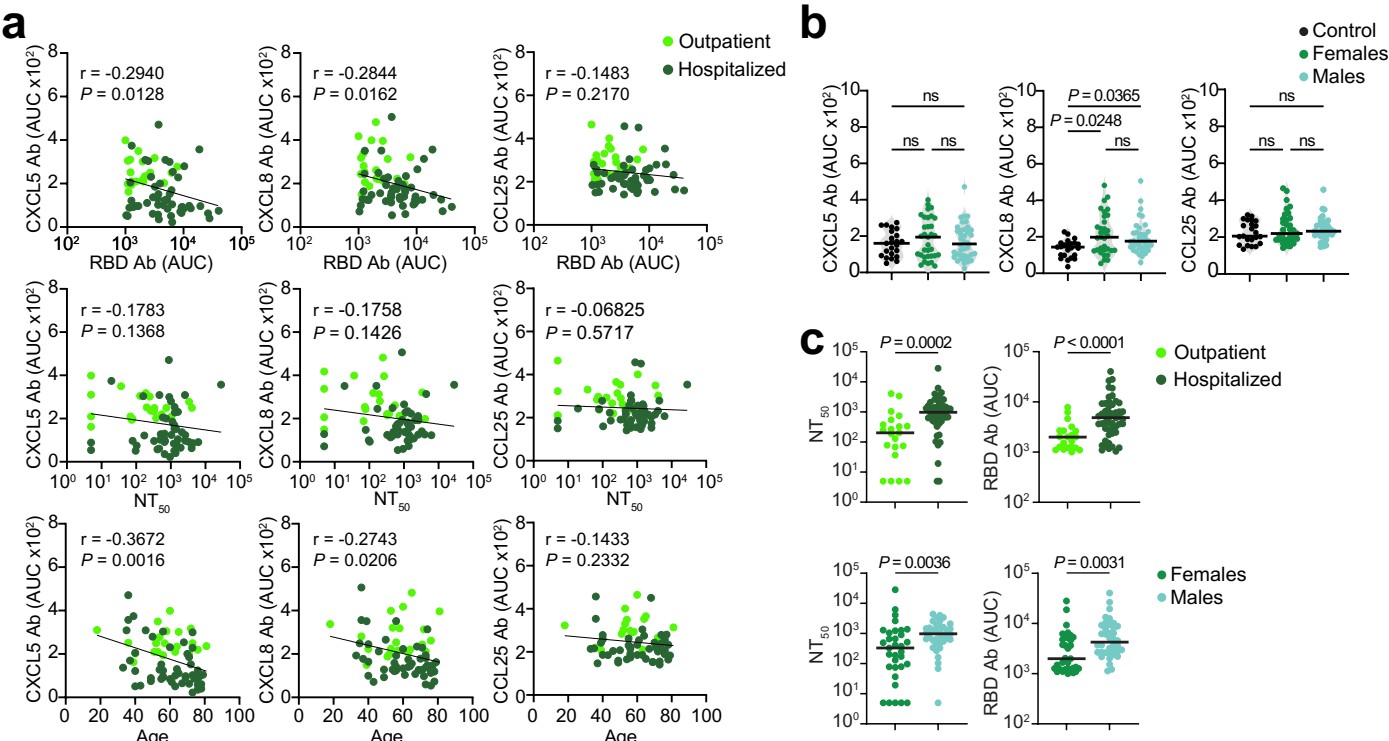

**Extended Data Fig. 6 | Correlation analyses of CXCL5, CXCL8 and CCL25 antibodies in COVID-19 convalescents.** (**a**) Two-tailed Pearson correlations of CXCL5, CXCL8 and CCL25 antibodies with RBD IgG, $NT_{50}$ values and age for COVID-19 convalescents from the Lugano cohort at month 6 that were previously hospitalized (n = 50) or outpatient (n = 21). Average of two independent experiments. (**b**) AUC of ELISA showing CXCL5, CXCL8 and CCL25 antibodies in healthy controls (Control, n = 23) and in COVID-19 convalescents (n = 71) from the Lugano cohort at month 6 grouped by gender (n = 33, females; n = 38, males). Average of two independent experiments. Horizontal bars indicate median values. Kruskal-Wallis test followed by Dunn's multiple comparison test. (**c**) AUC of ELISA showing $NT_{50}$ and RBD IgG values in COVID-19 convalescents (n = 71) from the Lugano cohort at month 6 grouped by disease severity (n = 50 hospitalized; n = 21, outpatient) and by gender (n = 33, females; n = 38, males). Average of two independent experiments. Horizontal bars indicate median values. Two-tailed Mann–Whitney U-tests.

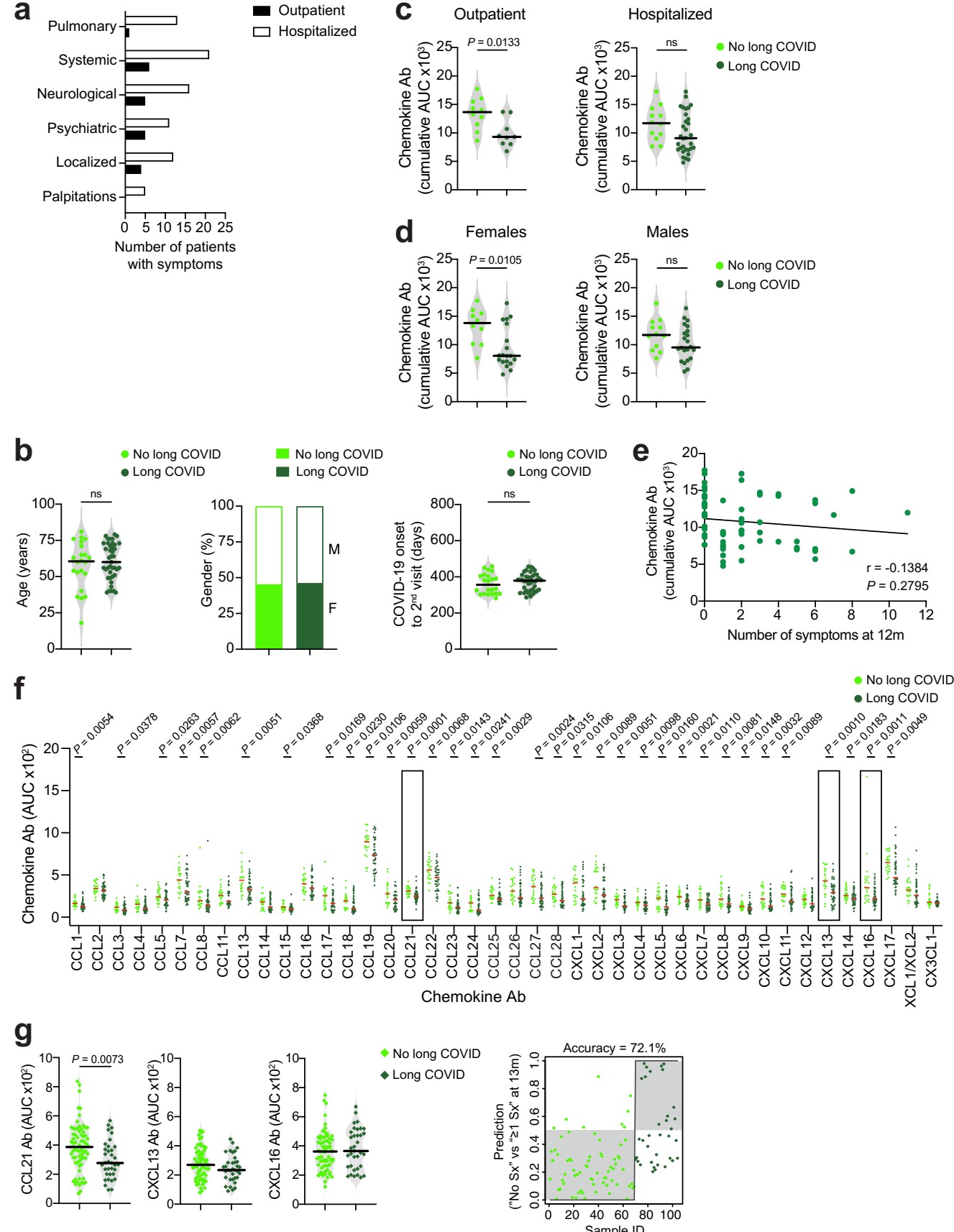

**Extended Data Fig. 7 | See next page for caption.**

**Extended Data Fig. 7 | Chemokine antibodies and long-term COVID-19 symptoms.** (**a**) Incidence of symptoms in patients with long COVID from the Lugano cohort at month 12 (n = 32, previously hospitalized; n = 9 outpatient). (**b**) Analysis of age (left), gender distribution (middle) and time from COVID-19 onset to month 12 sample collection (right) in COVID-19 convalescents in the Lugano cohort without (n = 22) and with long COVID (n = 41). Horizontal bars indicate median values. Two-tailed Mann–Whitney U-tests. (**c,d**) Cumulative AUC of ELISA showing chemokine IgG antibodies in COVID-19 convalescents without and with long COVID at month 12 in the Lugano cohort grouped by disease severity (c; n = 10, outpatient no long COVID; n = 9, outpatient long COVID; n = 12, hospitalized no long COVID; n = 32, hospitalized long COVID)) or by gender (d; n = 10, females no long COVID; n = 19, females long COVID; n = 12, males no long COVID; n = 22, males long COVID). Average of two independent experiments.

Horizontal bars indicate median values. Two-tailed Mann–Whitney U-tests. (**e**) Two-tailed Pearson correlation of the cumulative signal of the antibodies against the 42 chemokines in COVID-19 convalescents from the Lugano cohort at month 6 (n = 63) and the number of their self-reported symptoms at month 12. Average of two independent experiments. (**f**) AUC of ELISA showing chemokine antibodies at month 6 in no long COVID (n = 22) and long COVID (n = 41) groups from the Lugano cohort. Data are shown as average AUC of two independent experiments. Horizontal bars indicate median values. Two-tailed Mann–Whitney U-tests. (**g**) Left, AUC of ELISA showing CCL21, CXCL13 and CXCL16 antibodies in no long COVID (n = 69) and long COVID (n = 35) groups from the Zurich cohort at month 13. Two-tailed Mann–Whitney U-tests. Right, logistic regression analysis assignment of no long COVID and long COVID groups from the Zurich cohort based on CCL21, CXCL13 and CXCL16 antibodies at month 13.

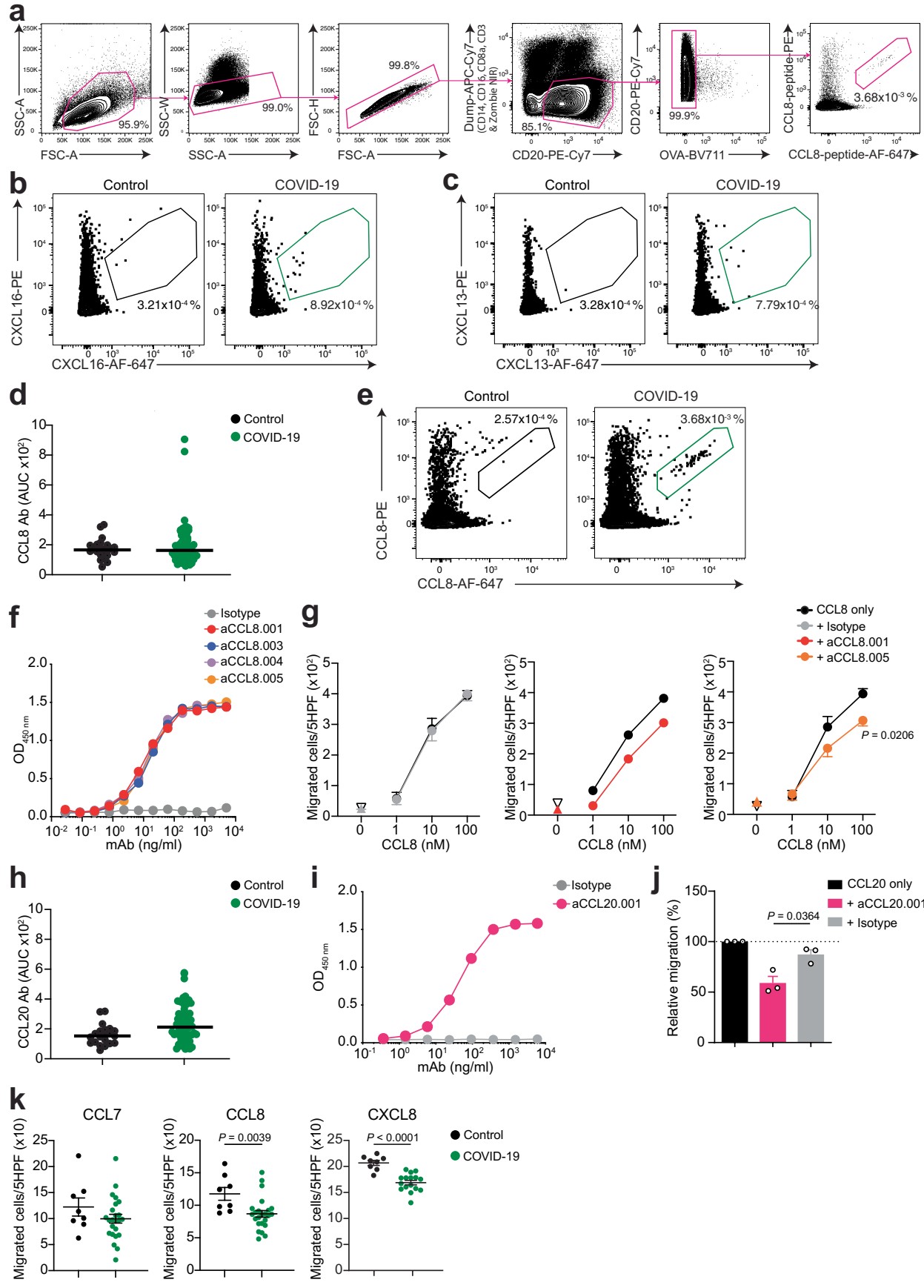

**Extended Data Fig. 8 | See next page for caption.**

**Extended Data Fig. 8 | Human monoclonal antibodies that impede chemotaxis.** (**a**) Gating strategy for sorting CCL8 N-loop specific B cells by flow cytometry. (**b**,**c**) Representative flow cytometry plots showing human B cells binding to the CXCL16 (b) or CXCL13 (c) N-loop peptide (gate). The frequency of antigen-specific B cells is shown. (**d**) AUC of ELISA showing antibodies to the CCL8 N-loop in healthy controls (Control, n = 23) and COVID-19 convalescents from the Lugano cohort at month 6 (COVID-19, n = 71), and identification of individuals with high antibody reactivity. Average of two independent experiments. Horizontal bars indicate median values. (**e**) Representative flow cytometry plots showing human B cells binding to the CCL8 N-loop peptide (gate). The frequency of antigen-specific B cells is shown. (**f**) ELISA showing CCL8 monoclonal antibodies binding to the CCL8 N-loop. Average of two independent experiments. (**g**) Chemotaxis showing migration of human monocytes in a CCL8 gradient (n = 4) in the presence of aCCL8.001 (n = 2), aCCL8.005 (n = 4) or isotype control antibody (n = 4). Mean±SEM of migrated cells in 5 high-power fields (HPF). Up-pointing triangle is antibody alone, and down-pointing triangle is buffer control. Two-way RM ANOVA followed by Šídák's multiple comparisons test. (**h**) AUC of ELISA showing antibodies to the CCL20 N-loop in healthy controls (n = 23) and COVID-19 convalescents from the Lugano cohort at month 6 (n = 71), and identification of individuals with high antibody reactivity. Average of two independent experiments. Horizontal bars indicate median values. (**i**) ELISA showing CCL20 monoclonal antibodies binding to the CCL20 N-loop. Average of two independent experiments. (**j**) Chemotaxis showing relative cell migration of the 300.19 preB cell line uniquely expressing CCR6 in a CCL20 gradient (1 nM) in the presence of aCCL20.001 or isotype control antibody. Mean+SEM of 3 independent experiments. Two-tailed Mann–Whitney U-tests. (**k**) Chemotaxis showing cell migration of preB 300.19 cells expressing CCR2 toward a CCL7 (100 nM) or CCL8 (100 nM) gradient, or of preB 300.19 cells expressing CXCR1 towards a CXCL8 gradient (1 nM), in the presence of plasma IgGs from a subset of COVID-19 convalescents from the Lugano cohort at month 6 (n = 24 for CCL7 and CCL8; n = 16 for CXCL8) or healthy controls (n = 8). Technical triplicates (Mean±SEM) of migrated cells in 5 high-power fields (HPF). Two-tailed Mann–Whitney U-tests.

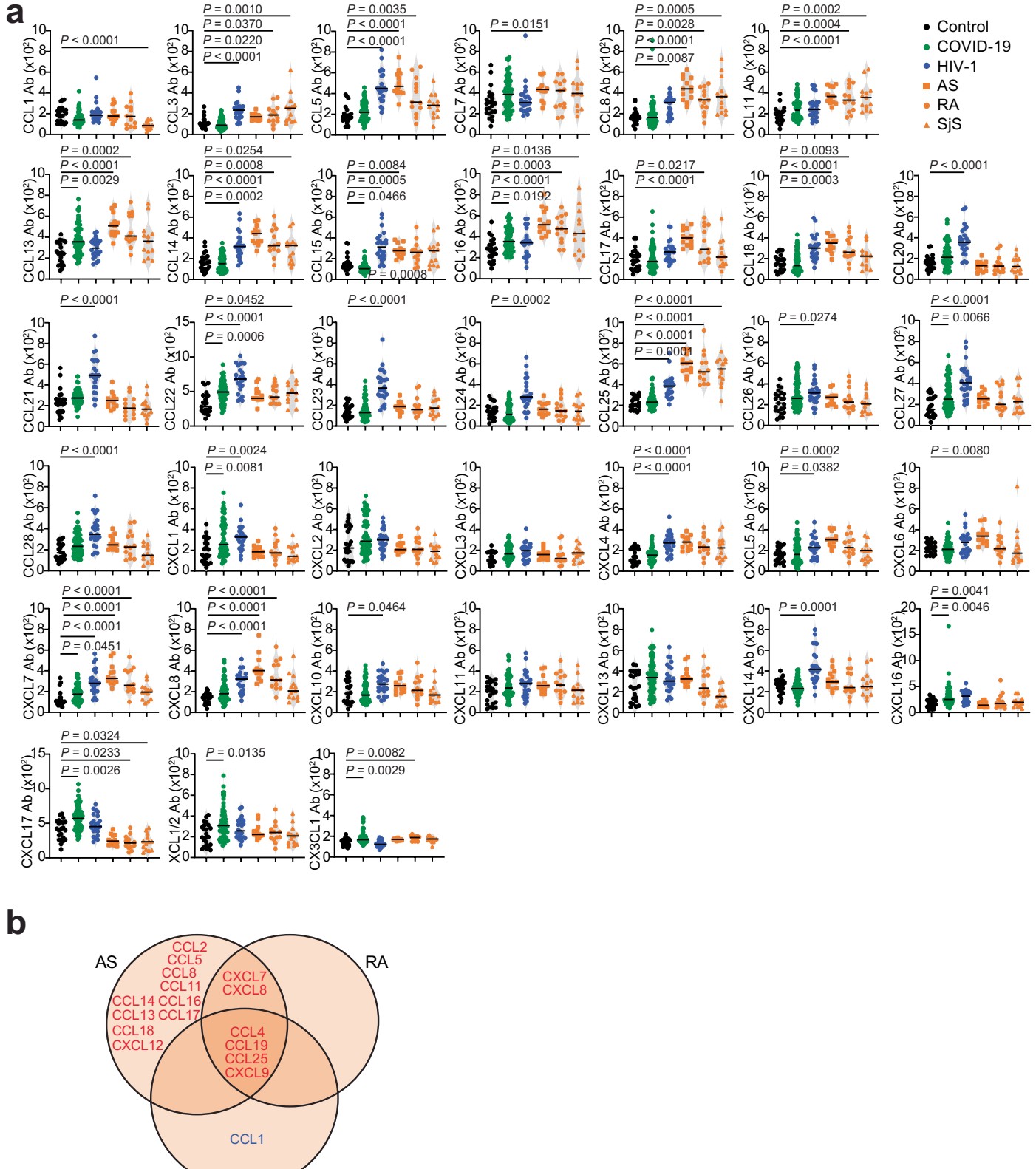

**Extended Data Fig. 9 | Chemokine antibodies in HIV-1 infection and in AS, RA and SjS.** (**a**) AUC of ELISA showing chemokine antibodies in healthy controls (Control, n = 23), COVID-19 convalescents (COVID-19, n = 71; Lugano cohort at month 6), HIV-1 (n = 24) AS (n = 13), RA (n = 13), and SjS (n = 13). Average of two independent experiments. Horizontal bars indicate median values. Statistical significance was determined using Kruskal-Wallis test followed by Dunn's multiple comparison test over rank of healthy controls. (**b**) Venn diagram showing the chemokines targeted by autoantibodies across the autoimmune disorders AS, RA and SjS. Red and blue colors indicate either an increase or decrease compared to healthy controls with $P < 10^{-4}$. Kruskal-Wallis test followed by Dunn's multiple comparison test over rank of healthy controls as in (a).

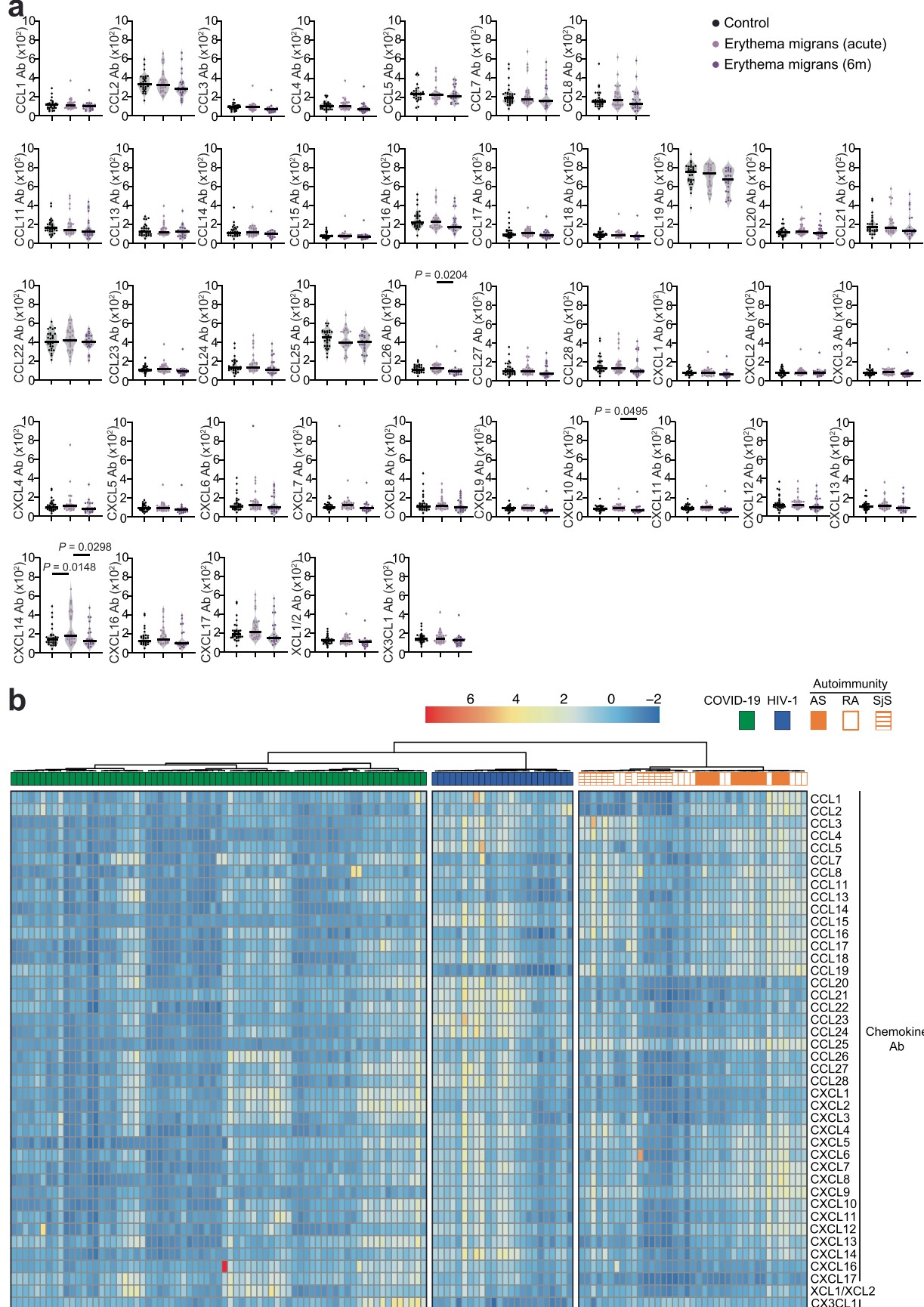

**Extended Data Fig. 10 | See next page for caption.**

**Extended Data Fig. 10 | Chemokine antibodies in Lyme disease (*Borrelia* infection) and clustering of COVID-19, HIV-1 and autoimmune diseases based on chemokine antibodies. (a**) AUC of ELISA showing chemokine antibodies in healthy controls (Control, n = 30) and in Lyme (Erythema migrans) during acute disease (n = 26) and at month 6 post-infection (n = 23). Average of two independent experiments. Horizontal bars indicate median values. Statistical significance was determined using Kruskal-Wallis test followed by Dunn's multiple comparison test. (**b**) Heatmap representing the unsupervised hierarchical clustering analysis of COVID-19 convalescents (n = 71; Lugano cohort at month 6), HIV-1 (n = 24), AS (n = 13), RA (n = 13) and SjS (n = 13), based on normalized AUC of ELISA values for plasma IgG binding to 42 peptides comprising the N-loop of all 43 human chemokines. The distribution of the groups within each cluster is shown.

# Reporting Summary

## Statistics

For all statistical analyses, confirm that the following items are present in the figure legend, table legend, main text, or Methods section.

| n/a | Confirmed | |
|---|---|---|
| ☐ | ☒ | The exact sample size (*n*) for each experimental group/condition, given as a discrete number and unit of measurement |
| ☐ | ☒ | A statement on whether measurements were taken from distinct samples or whether the same sample was measured repeatedly |
| ☐ | ☒ | The statistical test(s) used AND whether they are one- or two-sided *Only common tests should be described solely by name; describe more complex techniques in the Methods section.* |
| ☒ | ☐ | A description of all covariates tested |
| ☐ | ☒ | A description of any assumptions or corrections, such as tests of normality and adjustment for multiple comparisons |
| ☐ | ☒ | A full description of the statistical parameters including central tendency (e.g. means) or other basic estimates (e.g. regression coefficient) AND variation (e.g. standard deviation) or associated estimates of uncertainty (e.g. confidence intervals) |
| ☐ | ☒ | For null hypothesis testing, the test statistic (e.g. *F*, *t*, *r*) with confidence intervals, effect sizes, degrees of freedom and *P* value noted *Give P values as exact values whenever suitable.* |
| ☒ | ☐ | For Bayesian analysis, information on the choice of priors and Markov chain Monte Carlo settings |
| ☒ | ☐ | For hierarchical and complex designs, identification of the appropriate level for tests and full reporting of outcomes |
| ☐ | ☒ | Estimates of effect sizes (e.g. Cohen's *d*, Pearson's *r*), indicating how they were calculated |

*Our web collection on statistics for biologists contains articles on many of the points above.*

## Software and code

Policy information about availability of computer code

| | |
|---|---|
| Data collection | - Flow cytometry data were collected with the FACS Aria III instrument.<br>- ELISA data were obtained with an ELISA microplate reader (from BioTek) with the Gen5 software.<br>- Nanoluc Luciferase activity in cell lysates was measured using the Nano-Glo Luciferase Assay System (Promega) with Modulus II Microplate Reader User interface (TURNER BioSystems). |
| Data analysis | For data analysis, the following software were used:<br>- FlowJo Software (version 10.7.1)<br>- Gen5 Software (version 3.12; Agilent)<br>- Glomax software (version 3.2.3; Promega)<br>- IgPipeline (as described in Robbiani et al., 2020)<br>- Prism 9 (version 9.0.2; GraphPad Software)<br>- Pretty Heatmaps (pheatmap) R package v 1.0.12<br>- PyMOL 2.5.0 (Schrödinger, Inc.)<br>- Rtsne R package v 0.15<br>- R 4.1.1 (R Development Core Team)<br>- RStudio 2021.09.0 (RStudio)<br>- Adobe Illustrator 2023<br>- Code availability: https://github.com/cavallilab/chemopept |

For manuscripts utilizing custom algorithms or software that are central to the research but not yet described in published literature, software must be made available to editors and reviewers. We strongly encourage code deposition in a community repository (e.g. GitHub). See the Nature Portfolio guidelines for submitting code & software for further information.

# Data

Policy information about availability of data

All manuscripts must include a data availability statement. This statement should provide the following information, where applicable:
- Accession codes, unique identifiers, or web links for publicly available datasets
- A description of any restrictions on data availability
- For clinical datasets or third party data, please ensure that the statement adheres to our policy

All data analyzed during the present study are included in this article and its supporting information files. Source data are provided with this paper. PDB accession codes are 5T1A, 6MEO, 6WWZ, and 1ESR.

# Human research participants

Policy information about studies involving human research participants and Sex and Gender in Research.

**Reporting on sex and gender**

Sex/gender was not considered for the study design, which was based on sample availability. Data were however analyzed to determine whether sex/gender were confounders. The main findings described in this study apply to all Sex/genders.

**Population characteristics**

All the demographics and clinical characteristics of the participants can be found in Supplementary Table 1.

**Recruitment**

COVID-19 original cohort (Lugano):
71 participants, diagnosed with COVID-19 at the Clinica Luganese Moncucco (CLM, Switzerland) between 08.03.2020 and 22.11.2020, were enrolled in the study and divided into two groups, according to the severity of the acute disease. The hospitalized group included 50 participants; the outpatient group included 21 close contacts of the hospitalized group, who only received at-home care. Inclusion criteria for the hospitalized group were a SARS-CoV-2 positive nasopharyngeal swab test by real-time reverse transcription-polymerase chain reaction (RT PCR) and age ≥18 years. Inclusion criteria for the outpatient group were being a symptomatic close contact (living in the same household) of an individual enrolled in the hospitalized group and age ≥18 years. Serologic tests confirmed COVID-19 positivity for all the participants (Fig. 1a; Extended Data Fig. 4a). At the 12-month visit, participants were asked to indicate the presence or absence of persisting symptoms related to COVID-19 according to a questionnaire. The study was performed in compliance with all relevant ethical regulations and the study protocols were approved by the Ethical Committee of the Canton Ticino (ECCT): CE-3428 and CE 3960.

COVID-19 validation cohort 1 (Milan):
44 participants, diagnosed with COVID-19 and hospitalized at the Humanitas Research Hospital (Milan, Italy) between 10.03.2020 and 29.03.2021, were enrolled in the study. Inclusion criteria for the participants were a SARS-CoV-2 positive nasopharyngeal swab test by RT-PCR and age ≥18 years. Serologic tests confirmed COVID-19 positivity for the participants who were not tested by RT-PCR. The study was performed in compliance with all relevant ethical regulations and the study protocols were approved by the Ethical Committee of Humanitas Research Hospital (authorization n° 738/20 and n° 956/20).

COVID-19 validation cohort 2 (Zurich):
104 participants, diagnosed with COVID-19 at the University Hospital Zurich, the City Hospital Triemli Zurich, the Limmattal Hospital or the Uster Hospital between April 2020 and April 2021, were included in the study and divided into two groups, according to the severity of the acute disease. The hospitalized group included 38 participants, whereas the outpatient group included 66 individuals, who only received at-home care. Inclusion criteria for the participants were a SARS-CoV-2 positive nasopharyngeal swab test by RT-PCR and age ≥18 years. At the 13-month visit, blood was collected, and participants were asked by trained study physicians to indicate the presence or absence of persisting symptoms related to COVID-19. The study was performed in compliance with all relevant ethical regulations and the study protocols were approved by the Cantonal Ethics Committee of Zurich (BASEC #2016-01440).

Control cohort:
15 adult participants (≥18 years) with self-reported absence of prior SARS CoV-2 infection or vaccination (confirmed by negative serologic test, fig. S3A) were enrolled between November 2020 and June 2021. Additional 8 pre-pandemic samples were obtained from blood bank donors (ECCT: CE-3428).

Vaccination cohort:
16 adult participants (≥18 years) with self-reported absence of prior SARS-CoV-2 infection (confirmed by negative serologic test, fig. S4F) and who received two doses of mRNA-based COVID-19 vaccine 62, 63, were enrolled on the day of first vaccine dose or earlier, between November 2020 and October 2021 (ECCT: CE-3428).

HIV-1 and autoimmune diseases cohorts:
Pre-pandemic plasma samples were obtained from the following participants: 24 HIV-1 positive (ECCT: CE-813) 64, 13 each with Ankylosing Spondylitis, Rheumatoid Arthritis (ECCT: CE 3065, and Ethical Committee of the Canton Zurich EK-515), or Sjögren's syndrome (IRCCS Policlinico San Matteo Foundation Ethics Committee n.20070001302).

Lyme disease cohort:
Plasma samples of 27 individuals with erythema migrans (Lyme disease) and 30 controls were obtained at The Valley Hospital (Ridgewood, NJ, USA) and Lyme & Tick-borne Disease Research Center at Columbia University Irving Medical Center (New York, NY, USA) between 2015 and 2019. All were between 18-89 years of age and all denied being immunocompromised.
Lyme disease cohort: Individuals had new or recent onset erythema migrans, exposure to a Lyme endemic area in the prior

| Ethics oversight | 30 days and received no more than 3 weeks of antibiotic treatment. Healthy control cohort: Individuals reported being medically healthy, had an unremarkable physical exam and blood tests, had no signs or symptoms of infection or illness, denied having had a diagnosis and/or treatment for Lyme and/or another tick-borne disease within the past 5 years, and denied having a tick bite in the prior 6 months. The Lyme cohort samples were collected at the time of the erythema migrans and 6 months later in average. The study was performed in compliance with all relevant ethical regulations and the study protocol was approved by the New York State Psychiatric Institute Institutional Review Board (#6805). |
|---|---|
| | COVID-19 original cohort (Lugano): The study was performed in compliance with all relevant ethical regulations and the study protocols were approved by the Ethical Committee of the Canton Ticino (ECCT): CE-3428 and CE 3960.

COVID-19 validation cohort 1 (Milan): The study was performed in compliance with all relevant ethical regulations and the study protocols were approved by the Ethical Committee of Humanitas Research Hospital (authorization n° 738/20 and n° 956/20).

COVID-19 validation cohort 2 (Zurich): The study was performed in compliance with all relevant ethical regulations and the study protocols were approved by the Cantonal Ethics Committee of Zurich (BASEC #2016-01440).

Control cohort: ECCT: CE-3428.

Vaccination cohort: ECCT: CE-3428

HIV-1 and autoimmune diseases cohorts: HIV-1 (ECCT: CE-813); Ankylosing Spondylitis and Rheumatoid Arthritis (ECCT: CE 3065, and Ethical Committee of the Canton Zurich EK-515); and Sjögren's syndrome (IRCCS Policlinico San Matteo Foundation Ethics Committee n.20070001302).

Lyme disease cohort: The study was performed in compliance with all relevant ethical regulations and the study protocol was approved by the New York State Psychiatric Institute Institutional Review Board (#6805).

Written informed consent was obtained from all participants, and all samples were coded to remove identifiers at the time of blood withdrawal. Demographic, clinical, and serological features are reported in supplementary tables. |

Note that full information on the approval of the study protocol must also be provided in the manuscript.

# Field-specific reporting

Please select the one below that is the best fit for your research. If you are not sure, read the appropriate sections before making your selection.

☒ Life sciences ☐ Behavioural & social sciences ☐ Ecological, evolutionary & environmental sciences

For a reference copy of the document with all sections, see nature.com/documents/nr-reporting-summary-flat.pdf

# Life sciences study design

All studies must disclose on these points even when the disclosure is negative.

| Sample size | No tests were used to determine sample size. Sample size was determined by the number of samples that were available to us and that fit the criteria of the study. |
|---|---|
| Data exclusions | No data were excluded. |
| Replication | All results were performed in at least two or three independent experiments as described in each figure legend. |
| Randomization | Randomization does not apply to this study. Experimental groups were determined by disease status (control vs COVID-19), disease severity (outpatient vs hospitalized COVID-19 individuals), vaccination status, patient clinical outcome (development of Long COVID or not), and disease groups (COVID-19 vs HIV-1 vs autoimmunity). |
| Blinding | The operator was blind to the assignment of a patient sample to disease group. |

# Reporting for specific materials, systems and methods

We require information from authors about some types of materials, experimental systems and methods used in many studies. Here, indicate whether each material, system or method listed is relevant to your study. If you are not sure if a list item applies to your research, read the appropriate section before selecting a response.

## Materials & experimental systems

| n/a | Involved in the study |
|---|---|
| ☐ | ☒ Antibodies |
| ☐ | ☒ Eukaryotic cell lines |
| ☒ | ☐ Palaeontology and archaeology |
| ☒ | ☐ Animals and other organisms |
| ☒ | ☐ Clinical data |
| ☒ | ☐ Dual use research of concern |

## Methods

| n/a | Involved in the study |
|---|---|
| ☒ | ☐ ChIP-seq |
| ☐ | ☒ Flow cytometry |
| ☒ | ☐ MRI-based neuroimaging |

# Antibodies

**Antibodies used**

Commercially available antibodies:
- Anti-human CD14, APC-eFluor780, clone 61D3 (Thermo Fisher Scientific; Cat#47-0149-42; RRID:AB_1834358; dilution 1:200)
- Anti-human CD16, APC-eFluor780, clone eBioCB16 (CB16; Thermo Fisher Scientific; Cat#47-0168-41; RRID:AB_11219083; dilution 1:200)
- Anti-human CD20, PE-Cy7, clone L27 (IVD; BD Biosciences; Cat#335828; RRID:AB_2868689; dilution 1:200)
- Anti-human CD3, APC-eFluor780, clone OKT3 (Thermo Fisher Scientific; Cat#47-0037-41; RRID:AB_2573935; dilution 1:200)
- Anti-human CD8a, APC-eFluor780, clone OKT8 (Thermo Fisher Scientific; Cat#47-0086-42; RRID:AB_2573945; dilution 1:200)
- Anti-human IgG, HRP-linked whole Ab (GE Healthcare; Cat# NA933; RRID:AB_772208; dilution 1:5000)

Monoclonal antibodies produced in this study: aCCL8.001, aCCL8.003, aCCL8.004, aCCL8.005, aCCL20.001, aCXCL13.001, aCXCL13.002, aCXCL13.003, aCXCL16.001, aCXCL16.002 and aCXCL16.003

Isotype control used in this study: Z021 (see Robbiani et al., 2017 as described in the method section).

**Validation**

All the commercially available antibodies utilized in this study have been validated by the manufacturer. Details can be found on their websites:
- Anti-human CD14, APC-eFluor780, clone 61D3 (Thermo Fisher Scientific; Cat#47-0149-42; RRID:AB_1834358; dilution 1:200)
  Website: https://www.thermofisher.com/antibody/product/CD14-Antibody-clone-61D3-Monoclonal/47-0149-42
- Anti-human CD16, APC-eFluor780, clone eBioCB16 (CB16; Thermo Fisher Scientific; Cat#47-0168-41; RRID:AB_11219083; dilution 1:200)
  Website: https://www.thermofisher.com/antibody/product/CD16-Antibody-clone-eBioCB16-CB16-Monoclonal/47-0168-41
- Anti-human CD20, PE-Cy7, clone L27 (IVD; BD Biosciences; Cat#335828; RRID:AB_2868689; dilution 1:200)
  Website: https://www.bdbiosciences.com/en-ch/products/reagents/flow-cytometry-reagents/clinical-diagnostics/single-color-antibodies-asr-ivd-ce-ivd/cd20-pe-cy-7.335828
- Anti-human CD3, APC-eFluor780, clone OKT3 (Thermo Fisher Scientific; Cat#47-0037-41; RRID:AB_2573935; dilution 1:200)
  Website: https://www.thermofisher.com/antibody/product/CD3-Antibody-clone-OKT3-Monoclonal/47-0037-41
- Anti-human CD8a, APC-eFluor780, clone OKT8 (Thermo Fisher Scientific; Cat#47-0086-42; RRID:AB_2573945; dilution 1:200)
  Website: https://www.thermofisher.com/antibody/product/CD8a-Antibody-clone-OKT8-OKT-8-Monoclonal/47-0086-42
- Anti-human IgG, HRP-linked whole Ab (GE Healthcare; Cat# NA933; RRID:AB_772208; dilution 1:5000)
  Website: https://www.fishersci.se/shop/products/anti-human-igg-peroxidase-linked-species-specific-whole-antibody-from-sheep-secondary-antibody-ge-healthcare/10547065

The specificity of the monoclonal antibodies produced in this study has been validated by testing binding to their cognate human antigen in ELISA (Fig. 2d and 2f; Extended Data Fig. 8f and 8i) and by assessing the inhibition of cellular migration toward human chemokines (2e and 2g; Extended Data Fig. 8g and 8j) compared to an isotype control.

# Eukaryotic cell lines

Policy information about cell lines and Sex and Gender in Research

**Cell line source(s)**

- 293T (ACE2)
- HEK293T (ATCC CRL-11268)
- Expi293F
- PreB 300.19 murine cell line expressing hCCR2 (Ogilvie at al, 2001)
- PreB 300.19 murine cell line expressing hCCR6 (Ogilvie at al, 2001)
- PreB 300.19 murine cell line expressing hCXCR1 (Zaslaver at al, 2001)
- PreB 300.19 murine cell line expressing hCXCR6 (Loetscher at al, 1997)

**Authentication**

No authentication was performed for the commercially available cell lines.
Expression of the cognate chemokine receptor on PreB 300.19 murine cell lines was confirmed by flow cytometry, and selective activity of the human agonist assessed by in vitro migration.

**Mycoplasma contamination**

Cell lines were not tested for Mycoplasma.

**Commonly misidentified lines**
(See ICLAC register)

No commonly misidentified cell lines were used.

# Flow Cytometry

## Plots

Confirm that:

☒ The axis labels state the marker and fluorochrome used (e.g. CD4-FITC).

☒ The axis scales are clearly visible. Include numbers along axes only for bottom left plot of group (a 'group' is an analysis of identical markers).

☒ All plots are contour plots with outliers or pseudocolor plots.

☒ A numerical value for number of cells or percentage (with statistics) is provided.

## Methodology

| | |
|---|---|
| Sample preparation | B cells were enriched from PBMCs of uninfected controls or of COVID-19 convalescent individuals 6 months after COVID-19, using the pan-B-cell isolation kit according to manufacturer's instructions (Miltenyi Biotec, 130-101-638). The enriched B cells were subsequently stained in FACS buffer (PBS + 2% FCS + 1mM EDTA) with the following antibodies/reagents (all 1:200 diluted) for 30 min on ice: anti-CD20-PE-Cy7 (BD Biosciences, 335828), anti-CD14-APC-eFluor 780 (Thermo Fischer Scientific, 47-0149-42), anti-CD16-APC-eFluor 780 (Thermo Fischer Scientific, 47-0168-41), anti-CD3-APC-eFluor 780 (Thermo Fischer Scientific, 47-0037-41), anti-CD8-APC-eFluor 780 (Invitrogen, 47-0086-42), Zombie NIR (BioLegend, 423105), as well as fluorophore-labeled ovalbumin (Ova) and N-loop peptides. Live single Zombie-NIR−CD14−CD16−CD3−CD8−CD20+Ova−N-loop-PE+N-loop-AF647+ B cells were single-cell sorted into 96-well plates containing 4 µl of lysis buffer (0.5× PBS, 10 mM DTT, 3,000 units/ml RNasin Ribonuclease Inhibitors [Promega, N2615]) per well using a FACS Aria III, and the analysis was performed with FlowJo software. The sorted cells were frozen on dry ice and stored at −80 °C. |
| Instrument | Cell sorting was performed with the  FACS Aria III instrument. |
| Software | Data analysis was performed using the FlowJo Software (version 10.7.1; Three Star). |
| Cell population abundance | The percentage of each cell population shown in this study is depicted in each the FACS plot. |
| Gating strategy | Gating strategy is provided in the Supplementary Information. Briefly, all cells were gated on FSC-A/SSC-A, and doublets were eliminated using SSC-W/SSC-A and FSC-H/FSC-A gates. For live/dead discrimination, the Zombie NIR™ Fixable Viability Kit (BioLegend; Cat#423105) was utilized according to the manufacturer's instructions. B cells specific to a particular chemokine peptide were gated as CD14−CD16−CD3−CD8−CD20+Ova−N-loop-PE+N-loop-AF647+. |

☒ Tick this box to confirm that a figure exemplifying the gating strategy is provided in the Supplementary Information.

