## [Peer Review File · Nature Immunology]

Peer Review Information

Journal: Nature Immunology

Manuscript Title: Autoantibodies against chemokines post-SARS-CoV-2 infection correlate with disease course

Corresponding author name(s): Mariagrazia Ugucconi, Davide F. Robbiani, Andrea Cavalli

Reviewer Comments & Decisions:

Decision Letter, initial version:

Subject: Decision on Nature Immunology submission NI-LE34351

Message: 30th Aug 2022

Dear Dr. Robbiani,

Thank you for your response to the referees comments on your Letter, "Anti-chemokine antibodies after SARS-CoV-2 infection correlate with favorable disease course". Although we are interested in the possibility of publishing your study in Nature Immunology, the issues raised by the referees need to be addressed.

Please revise along the lines specified in your letter and please include the analysis of the cytokine expression data as requested by the referees. At resubmission, please include a "Response to referees" detailing, point-by-point, how you addressed each referee comment. If no action was taken to address a point, you must provide a compelling argument. This response will be sent back to the referees along with the revised manuscript.

Please include a revised version of any required reporting checklist. It will be available to referees to aid in their evaluation.

Reporting summary:

- that unprocessed scans are clearly labelled and match the gels and western blots presented in figures.
- that control panels for gels and western blots are appropriately described as loading on sample processing controls

-- all images in the paper are checked for duplication of panels and for splicing of gel lanes.

Please use the link below to submit your revised manuscript and related files:
[REDACTED]

We hope to receive your revised manuscript within three months. If you cannot send it within this time, please let us know. We will be happy to consider your revision so long as nothing similar has been accepted for publication at Nature Immunology or published elsewhere.

Nature Immunology is committed to improving transparency in authorship. As part of our efforts in this direction, we are now requesting that all authors identified as 'corresponding author' on published papers create and link their Open Researcher and Contributor Identifier (ORCID) with their account on the Manuscript Tracking System (MTS), prior to acceptance. ORCID helps the scientific community achieve unambiguous attribution of all scholarly contributions. You can create and link your ORCID from the home page of the MTS by clicking on 'Modify my Springer Nature account'. For more information please visit www.springernature.com/orcid.

Sincerely,

Ioana Visan, Ph.D.
Senior Editor
Nature Immunology

Tel: 212-726-9207
Fax: 212-696-9752
www.nature.com/ni

Reviewers' Comments:

Reviewer #1:

Remarks to the Author:

These authors devised a peptide-based ELISA strategy to identify and measure auto-antibodies reactive to functional domains of 43 human chemokines. Analysis of convalescent plasma from a COVID-19 infected cohort identified associations of anti-chemokine antibody levels with severity of acute COVID and development of long COVID. The authors claim that increased anti-chemokine antibodies found in individuals with favorable outcomes suggest a role of anti-chemokine antibodies in immune regulation.

The development of antibodies against cytokines and immune effector molecules has been described in COVID-19 and associate with adverse outcomes. This manuscript presents a large body of highly relevant data with clear focus on the original idea that anti-chemokine antibodies in COVID-19 may be protective against collateral immune mediated damage during COVID-19 infection. The sheer number of anti-chemokine autoantibodies that increase after COVID-19 is striking - 23 out of the 43 chemokines. However, high anti-chemokine antibody levels in controls are also striking and warrants discussion and explanation.

Strong correlation between antibody levels against N-loop and C-terminal epitopes provide confirmation of antibody reactivity against multiple chemokine epitopes. Generation of monoclonal antibodies from a subset of subjects and functional analysis in chemotaxis assays provides confirmation that the anti-chemokine antibodies are functionally active to inhibit immune cell migration and represents a major strength of this manuscript.

The observation of higher anti-chemokine antibodies in patients with favorable outcomes is provocative but should be interpreted cautiously given that this is an observational study. Problematic interpretation of observational data is a major problem with this study. Sources of bias need to be more rigorously investigated and alternative explanations need to be carefully considered. The alternative interpretation that antibody development to these chemokines may reflect higher production of these antibodies in lymphoid organs during infection needs to be considered because the implications of this interpretation are very different from what is stated in the abstract - "antibodies associated with favorable COVID-19 may be beneficial by modulating the inflammatory response and thus bear therapeutic potential." If the anti-chemokine antibodies associated with favorable COVID-19 outcome are a reflection of higher chemokine production, then these chemokines may actually play a protective role in COVID-19 immunity and disease resolution.

There are several areas for improvement.

1. Given these are observational human data, defining the study cohorts is essential. Tables should be provided with information about age, sex, race/ethnicity, and relevant comorbidities (hypertension, diabetes, obesity, immune deficiency) for both the COVID-19 subjects and the controls. How were the control subjects identified and confirmed to be COVID-19 naive?
2. The technical limitations of the peptide-based antibody assays should be defined. What is the lower limit of detection and how is it defined? Did the controls show evidence for chemokine auto-antibodies or were they undetectable? It seems from the data that anti-chemokine antibodies are common in controls and COVID-19 subjects with quantitative changes in the COVID-19 subjects between 0 and 1 logFC. If true, the high frequency of

autoreactivity against chemokines among healthy controls needs to be discussed and explained.

3. Statistical analysis assessing anti-chemokine antibody level as predictor of hospitalization or long COVID do not take into consideration important clinical and demographic subject covariates known to influence COVID-19 severity such as age, race/ethnicity, hypertension, diabetes, obesity, immune deficiency, etc... Models should address clinical and demographic variables likely to influence the outcome. Alternative explanations for the antibody-outcome associations must be considered to have a valid study.

4. The chemotaxis assay in Figure 2F shows inhibition of CXCL16 mediated migration by three of the monoclonals. The comparison for statistical analysis is made between untreated cells and the monoclonal antibody treated cells. Why was the direct comparison not made for the isotype control instead, given potential non-specific effects of antibody treatment on the migration assay?

5. Are the chemokine antibody concentrations present in subject sera at high enough levels that they would be expected to inhibit chemokine activity? Given that the antibodies are present in subject sera in a polyclonal mixture, experiments could be performed to address this question using COVID-19 convalescent and control sera in chemotaxis assays.

6. The authors do not address the question - why these chemokine autoantibodies are produced? Broad chemokine autoreactivity observed across autoimmune diseases, HIV and COVID-19 as well as detectable but lower levels in healthy controls suggests that chemokine autoantibodies may generally correlate with chemokine levels and not as a specific phenomenon during COVID-19 infection. The profile of chemokine antibodies in the COVID-19 subjects could be a reflection of the chemokine profile produced during the infection. Thus, the anti-chemokine antibody profile of subjects with clinically favorable course might reflect a profile of chemokines that contribute to favorable disease resolution. Given that sera are available from many of these subjects, it is very important to consider and quantify the relationship between chemokine antibodies and chemokine concentrations. Direct correlation of chemokine concentrations with antibody levels would provide evidence that antibody production is stimulated by chemokine production. Lack of association would help to dismiss this model. In addition, higher level of chemokine present in subjects with antibodies would be expected to counteract neutralizing activity of the anti-chemokine antibodies.

Reviewer #3:

Remarks to the Author:

The present study by Muri and colleagues investigates the concentrations of anti-chemokines antibodies in convalescent patients with COVID-19. They describe various anti-chemokine signatures differentiating between post-COVID-19 individuals and controls, between individuals that have been hospitalised or not in the acute phase of COVID-19, and those with long-term symptoms after the infectious episode. The question asked by this study is relevant, but important questions remain.

Comments

1. The major limitation of the study is that the cohorts studied are small. When the authors perform sub-analyses based on various criteria (hospitalisation or not, presence of long-term symptoms or not), the cohort of COVID-19 is split further in smaller groups.
2. Validation of the findings in an independent cohort is necessary to strengthen the conclusions.
3. The presence of completely different anti-chemokine antibody signatures for the various comparisons investigated (COVID-19 vs healthy, hospitalisation vs outpatient, long-term symptoms or not) is also puzzling. There is no consistent pathophysiological hypothesis to explain this.
4. The percentage of individuals with persistent symptoms is much higher than in other epidemiological studies. It is unclear which are the clinical criteria on which such persistence was based. There is no consistent diagnosis of long-COVID19 according to the very limited description.
5. The authors hypothesise in the Discussion that anti-chemokine antibodies may protect against severe COVID-19 disease. But the major differences were seen here 6 months later, not in the acute phase, how could they have protected?
6. The presence of anti-chemokine antibodies by this methodology does not seem to be specific for COVID-19, being present also in HIV infection. How much is likely that this is important for pathophysiology rather than an epiphenomenon?

Reviewer #4:

Remarks to the Author:

In this study, the authors discovered anti-chemokine Abs were associated with favorable COVID-19 and the lack of long COVID symptoms. In addition, COVID-19 specific anti-chemokine Ab profiles were unique and different HIV and autoimmune diseases. These Abs potentially bind to N-loop of chemokine and could reduce cell migration. In general, these findings are novel but lack of mechanisms.

1. This study found three groups of anti-chemokines Abs which potentially identified COVID-19, hospitalization and long COVID-19. The findings are interesting but lack of mechanisms. Although they perform several blocking assays using anti-chemokine Ab in vitro, it is insufficient to address the anti-chemokine Ab function given lack of in vivo data.
2. What is relationship between the anti-chemokines Ab and chemokines in plasma? The author should detect all of the chemokines and analyzed their levels, and performed the correlations with anti-chemokine Ab levels at early 0.5m, 6m and 12m. These data will provide more useful information of anti-chemokine Abs in clinic.
3. The authors collected samples at 6 month and 12 month and profiled anti-chemokine Abs. They found COVID-19 signatures relation to healthy donor, hospitalization signatures relation to outpatient and more important long COVID-19 signatures. However, COVID-19 and hospitalization signatures were present at 6 months after onset of symptom, which lack of prediction values. The authors should profile early sample to find the COVID-19 and hospitalization signatures because they have collected 0.5m samples.
4. Although this study suggests COVID-19 and hospitalization signatures, these anti-chemokine Abs lack of predict and therapeutic values because these signatures happen at 6 months after the onset of symptom rather than early phase of illness. By contrast, long COVID signatures are more important. Alternatively, I suggest Figure 1 and 3 moved into supplemental dataset and manuscript should be reduced into brief report which focus long COVID19 with one main figure.
5. Lack of vaccine incubation information, such as name, incubation time, et al.

Author Rebuttal to Initial comments**Point-by-point response to the Reviewers' comments**

We thank the Reviewers for carefully reading our manuscript and for the constructive criticism, which we addressed as described point-by-point in this response with new experiments, data analyses and changes to the text. Particularly, we examined chemokine levels at different time points after the infection and validated key discoveries with two additional and independent COVID-19 cohorts. Moreover, we performed chemotaxis assays in the presence of total plasma IgG, which demonstrates that polyclonal antibodies from COVID-19 convalescents can impair cell migration even at sub-physiologic concentrations, and analyzed autoantibodies in *Borrelia*-infected individuals (which were examined because Lyme disease can cause long-term symptoms similar to long COVID).

Reviewer #1

These authors devised a peptide-based ELISA strategy to identify and measure auto-antibodies reactive to functional domains of 43 human chemokines. Analysis of convalescent plasma from a COVID-19 infected cohort identified associations of anti-chemokine antibody levels with severity of acute COVID and development of long COVID. The authors claim that increased anti-chemokine antibodies found in individuals with favorable outcomes suggest a role of anti-chemokine antibodies in immune regulation.

The development of antibodies against cytokines and immune effector molecules has been described in COVID-19 and associate with adverse outcomes. This manuscript presents a large body of highly relevant data with clear focus on the original idea that anti-chemokine antibodies in COVID-19 may be protective against collateral immune mediated damage during COVID-19 infection. The sheer number of anti-chemokine autoantibodies that increase after COVID-19 is striking - 23 out of the 43 chemokines. However, high anti-chemokine antibody levels in controls are also striking and warrants discussion and explanation.

Strong correlation between antibody levels against N-loop and C-terminal epitopes provide confirmation of antibody reactivity against multiple chemokine epitopes. Generation of monoclonal antibodies from a subset of subjects and functional analysis in chemotaxis assays provides confirmation that the anti-chemokine antibodies are functionally active to inhibit immune cell migration and represents a major strength of this manuscript.

The observation of higher anti-chemokine antibodies in patients with favorable outcomes is provocative but should be interpreted cautiously given that this is an observational study. Problematic interpretation of observational data is a major problem with this study. Sources of bias need to be more rigorously investigated and alternative explanations need to be carefully considered. The alternative interpretation that antibody development to these chemokines may reflect higher production of these antibodies in lymphoid organs during infection needs to be considered because the implications of this interpretation are very different from what is

stated in the abstract - "antibodies associated with favorable COVID-19 may be beneficial by modulating the inflammatory response and thus bear therapeutic potential." If the anti-chemokine antibodies associated with favorable COVID-19 outcome are a reflection of higher chemokine production, then these chemokines may actually play a protective role in COVID-19 immunity and disease resolution.

There are several areas for improvement.

We thank Reviewer 1 for recognizing that "This manuscript presents a large body of highly relevant data with clear focus on the original idea that anti-chemokine antibodies in COVID-19 may be protective against collateral immune mediated damage during COVID-19 infection." and for stating that "Generation of monoclonal antibodies from a subset of subjects and functional analysis in chemotaxis assays provides confirmation that the anti-chemokine antibodies are functionally active to inhibit immune cell migration and represents a major strength of this manuscript."

1. *Given these are observational human data, defining the study cohorts is essential. Tables should be provided with information about age, sex, race/ethnicity, and relevant comorbidities (hypertension, diabetes, obesity, immune deficiency) for both the COVID-19 subjects and the controls. How were the control subjects identified and confirmed to be COVID-19 naive?*

→ We thank Reviewer 1 for pointing out the importance of providing complete demographic and clinical information for the COVID-19 study cohorts.

Supplementary Table 1 includes information about age, sex and relevant comorbidities for all COVID-19 convalescents (original Lugano cohort). The same information was added for the controls. Race/ethnicity was almost 100% Caucasian in both groups, and this information is now included. Serologic tests confirmed COVID-19 negativity for all controls (see Supplementary Table 1; Methods [lines 449-450]; Fig. 1a and Extended Data Fig. 4a in the revised manuscript).

Since we subsequently analyzed samples from two additional COVID-19 cohorts (from Zurich and Milan), the demographic and clinical information for these is also provided with the revised manuscript (see additional sheets in Supplementary Table 1 in the revised manuscript).

2. *The technical limitations of the peptide-based antibody assays should be defined. What is the lower limit of detection and how is it defined? Did the controls show evidence for chemokine auto-antibodies or were they undetectable? It seems from the data that anti-chemokine antibodies are common in controls and COVID-19*

subjects with quantitative changes in the COVID-19 subjects between 0 and 1 logFC. If true, the high frequency of autoreactivity against chemokines among healthy controls needs to be discussed and explained.

→ We thank the Reviewer for pointing out that for some anti-chemokine antibodies the signal in controls is above background, and that technical limitations of the assay should be discussed.

The peptide-based antibody assay cannot exclude the presence of autoantibodies prior to infection or in the control group. We address this point in the Discussion where we state that autoantibodies “*are detected early on during the acute phase, suggesting that they are either **pre-existing** or rapidly induced following the infection*” (lines 354-356). Thus, the presented values should be interpreted as relative rather than absolute.

As indicated in the Methods (line 507), “*an irrelevant peptide was used as negative control*” to determine the overall background of the assay and help to define the lower limit of detection (see Figure 1a, Extended Data Fig. 6a, and ‘negative control’ at the bottom and Supplementary Table 2 for control peptide sequence). However, the basal average optical density likely also depends on intrinsic features of each peptide that is used to coat the ELISA plate. We pointed out these technical limitations of the assay in the Methods (lines 571-574): “*Since the basal average optical density likely also depends on intrinsic features of each peptide that is used to coat the ELISA plate, the presented values should be interpreted as relative rather than absolute*”.

Unlike establishing an assay for the measurement of cytokines or other factors, which can be accomplished with the aid of a standard curve with known concentrations of the factor of interest, measuring the concentration of polyclonal antibodies is much complicated by the fact that they are a pool of heterogeneous molecules with different binding properties to the antigen of interest.

3. Statistical analysis assessing anti-chemokine antibody level as predictor of hospitalization or long COVID do not take into consideration important clinical and demographic subject covariates known to influence COVID-19 severity such as age, race/ethnicity, hypertension, diabetes, obesity, immune deficiency, etc... Models should address clinical and demographic variables likely to influence the outcome. Alternative explanations for the antibody-outcome associations must be considered to have a valid study.

→ We agree with the Reviewer that it is important to consider demographic and clinical covariates. For this reason, we had already analyzed gender and age as possible confounders in the first submission of the manuscript (now in Extended Data Fig. 4d,e; Extended Data Fig. 6c,d; and Extended Data Fig. 7c,e of the revision).

As suggested, we additionally performed χ^2 -tests considering covariates that are known to influence COVID-19 severity (demographics [gender and age] and comorbidities [diabetes and cardiovascular diseases]) and found that none of them was significantly different between groups. (Race/ethnicity was not analyzed because the cohort is nearly 100% Caucasian; similarly, immune deficiency was rare).

a COVID-19 severity
(outpatient vs hospitalized)

Covariate	p-value
Age	0.1252
Gender	0.1532
Diabetes	0.3198
Cardiovascular diseases	0.4261

b Long COVID
(no Sx vs ≥ 1 Sx)

Covariate	p-value
Age	0.3211
Gender	1
Diabetes	0.2231
Cardiovascular diseases	0.5064

Pearson's Chi-squared test with Yates' continuity correction in COVID-19 convalescents in the Lugano cohort were grouped based on COVID-19 severity (a) and long COVID (b).

Moreover, as expected based on the literature, the combination of these covariates (age, gender, diabetes and cardiovascular diseases) allows proper assignment with accuracies of 74.6% (COVID-19 severity; outpatient vs hospitalized) and 68.3% (Long COVID; no Sx vs ≥ 1 Sx). Notably, the accuracy using anti-chemokine antibody values is even better (77.5% [COVID-19 severity]) and 77.8% [Long COVID]).

Logistic regression Accuracy (%)	
Age + gender + diabetes + cardiovascular diseases	
Hospitalization signature	74.6
Long COVID signature	68.3
Anti-chemokine signature	
Hospitalization signature	77.5
Long COVID signature	77.8

Logistic regression analysis with covariates (age, gender, diabetes and cardiovascular disease) versus anti-chemokine antibody signatures.

The results of these new analyses are reported in the Methods section (lines 713-722) and in Supplementary Table 7.

4. The chemotaxis assay in Figure 2F shows inhibition of CXCL16 mediated migration by three of the monoclonals. The comparison for statistical analysis is made between untreated cells and the monoclonal antibody treated cells. Why was the direct comparison not made for the isotype control instead, given potential non-specific effects of antibody treatment on the migration assay?

→ We thank the Reviewer for catching this oversight. Clearly isotype is the appropriate comparison. The statistical analysis confirms significance by comparing the isotype control to either aCXCL16.001 ($p=0.030$) or aCXCL16.002 ($p=0.007$; paired-t-test):

Inhibition of chemotaxis by anti-CXCL16 N-loop antibodies.

The new analysis is presented in Fig. 2f and the text was modified accordingly.

5. Are the chemokine antibody concentrations present in subject sera at high enough levels that they would be expected to inhibit chemokine activity? Given that the antibodies are present in subject sera in a polyclonal mixture, experiments could be performed to address this question using COVID-19 convalescent and control sera in chemotaxis assays.

→ We thank the Reviewer for suggesting these experiments, which we performed.

We directly addressed this with new chemotaxis assays and found that plasma IgG from COVID-19 convalescents can indeed significantly inhibit migration of immune cells toward chemokines that attract inflammatory cells such as monocytes and neutrophils, even at concentrations below those present in human blood:

Polyclonal IgGs from COVID-19 convalescents inhibit chemotaxis.

The new data are presented in new Extended Data Fig. 8k and the Results and Methods sections were modified accordingly.

6. *The authors do not address the question - why these chemokine autoantibodies are produced? Broad chemokine autoreactivity observed across autoimmune diseases, HIV and COVID-19 as well as detectable but lower levels in healthy controls suggests that chemokine autoantibodies may generally correlate with chemokine levels and not as a specific phenomenon during COVID-19 infection. The profile of chemokine antibodies in the COVID-19 subjects could be a reflection of the chemokine profile produced during the infection. Thus, the anti-chemokine antibody profile of subjects with clinically favorable course might reflect a profile of chemokines that contribute to favorable disease resolution. Given that sera are available from many of these subjects, it is very important to consider and quantify the relationship between chemokine antibodies and chemokine concentrations. Direct correlation of chemokine concentrations with antibody levels would provide evidence that antibody production is stimulated by chemokine production. Lack of association would help to dismiss this model. In addition, higher level of chemokine present in subjects with antibodies would be expected to counteract neutralizing activity of the anti-chemokine antibodies.*

→ We thank the Reviewer for prompting us to measure chemokines and correlate with autoantibody levels, which we did.

We quantified chemokine concentrations at different time points in two independent cohorts and performed correlation analyses with the levels of the corresponding autoantibody in plasma. In agreement with published literature, several chemokines were significantly elevated during acute disease (including those related to the hospitalization signature) and many of them remained above baseline levels at 7 months (see Fig. 3a in the revised manuscript):

Plasma chemokine levels in the Milan (n=44) and Lugano (n=12) cohorts at the indicated time points after disease onset.

However, none of the chemokines corresponding to the hospitalization signature was significantly different between mild and severe patients. Accordingly, no correlation was observed between the levels of the chemokine and of the related autoantibody, neither in the acute phase nor at 7 months from disease onset (see Fig. 3b,c):

Levels of COVID-19 hospitalization signature chemokines (CXCL8, CCL25 and CXCL5) in mild versus severe patients (Milan cohort).

The new data are presented in new Fig. 3 and the text was modified accordingly.

Although the experimental data and correlation analyses dismiss the simple model that chemokines proportionally induce autoantibodies (the higher the chemokines, the higher the autoantibodies), we cannot exclude that the overall lack of correlation may be due to the timing of sampling and different half-life of antibodies and chemokines in plasma, or that plasma levels may not reflect chemokine concentrations in tissues, which could be more relevant for antibody induction.

Further highlighting the complexity of the phenomenon, autoantibodies are not induced against some of the chemokines that are remarkably increased during COVID-19 (e.g. CCL3, CCL4 and CXCL9, see above and Fig. 3a). These considerations were added to the text (lines 363-369).

Reviewer #3

The present study by Muri and colleagues investigates the concentrations of anti-chemokines antibodies in convalescent patients with COVID-19. They describe various anti-chemokine signatures differentiating between post-COVID-19 individuals and controls, between individuals that have been hospitalised or not in the acute phase of COVID-19, and those with long-term symptoms after the infectious episode. The question asked by this study is relevant, but important questions remain.

Comments

- 1. The major limitation of the study is that the cohorts studied are small. When the authors perform sub-analyses based on various criteria (hospitalisation or not, presence of long-term symptoms or not), the cohort of COVID-19 is split further in smaller groups.*
- 2. Validation of the findings in an independent cohort is necessary to strengthen the conclusions.*

Points 1 and 2 are linked by evidencing the small size of the cohort (Point 1) and asking for validation in an independent cohort (Point 2).

→ We thank Reviewer 3 for raising these issues. One could argue that statistically significant differences between groups, despite the modest size of the cohort, represent a strength rather than a weakness of the study. However, we agree that it is important to validate with independent COVID-19 cohorts, which is what we did by analyzing with the same methods samples from:

- a cohort from Milan (44 individuals) and
- a cohort from Zurich (104 individuals).

Like with the original cohort (71 individuals), both of the two additional cohorts were established during the first epidemic wave (before vaccines were available). For the Milan cohort, samples were collected at two different time points (acute and 7 months after symptoms onset) and data regarding severity of the acute disease were available (but not for long COVID). Samples from the Zurich cohort were collected at 13 months and data regarding the severity of the acute disease and persistence of long COVID at 13 months were available. The demographic and clinical data of the Milan and Zurich cohorts are summarized below and reported in Extended Data Fig. 3a and Supplementary Table 1:

Milan					Zurich				
Gender	n	Average age in years (range)	Severe/mild	% with symptoms at 12 months	Gender	n	Average age in years (range)	Hospitalized/outpatient	% with symptoms at 13 months (Yes/No)
Male	26	61 (40-90)	19/7	not known	Male	56	47 (20-84)	23/33	38% (21/35)
Female	18	65 (37-89)	12/6	not known	Female	48	44 (20-79)	15/33	29% (14/34)

Characteristics of the Milan and Zurich cohorts.

Importantly, the key findings from the original analysis were validated by the two additional cohorts, in particular:

COVID-19 signature (anti-CCL19, anti-CCL22 and anti-CXCL17):

The levels of the three COVID-19 signature anti-chemokine antibodies were all confirmed to be significantly increased in COVID-19 patients from the Milan (t=acute and t=7months) and Zurich (t=13months) cohorts. Accordingly, values from the 3 anti-chemokine antibodies combined properly assigned individuals to COVID-19 or control groups with accuracies of 90.5% (t=acute; Milan cohort), 89.5% (t=7months; Milan cohort) and 92.9% (t=13months; Zurich cohort). The accuracy in the original cohort is 96.8% at 6 months.

The new data are reported in Extended Data Fig. 3b, and the text was modified accordingly:

Validation of the COVID-19 signature in the Milan and Zurich cohorts.

Hospitalization signature (anti-CXCL8, anti-CCL25, and anti-CXCL5):

The three hospitalization signature anti-chemokines antibodies were overall higher in mild/outpatient over severe/hospitalized individuals in both additional cohorts. Accordingly, values from the 3 anti-chemokine antibodies combined properly assigned individuals to their respective group with accuracies of 85.0% (t=acute; Milan cohort), 84.1% (t=7months; Milan cohort) and 73.1% (t=13months; Zurich cohort). The accuracy in the original cohort is 77.5% at 6 months.

The new data are reported in Extended Data Fig. 3c, and the text was modified accordingly:

Validation of the hospitalization signature in the Milan and Zurich cohorts.

Note: anti-CCL2 antibodies were previously included as part of the hospitalization signature. However, since we did not observe significant differences for anti-CCL2 antibodies in the two validation cohorts, we decided to remove anti-CCL2 from the hospitalization signature also in the original Lugano cohort.

Long COVID signature (anti-CCL21, anti-CXCL13 and anti-CXCL16):

Analysis of the Zurich cohort at 13 months showed 72.1% accuracy of association with lack of long COVID, even though in that cohort only anti-CCL21 antibodies were significantly different between groups.

With this regard, we note that, unlike anti-CCL21 antibodies, those against CXCL13 and CXCL16 significantly decreased from 6 to 12 months in the original cohort (Extended Data Fig. 5c), which might explain why only

autoantibodies against CCL21 remained significantly higher at 13 months in individuals without persisting symptoms in the Zurich cohort. The accuracy of prediction in the original cohort is 77.8% at 6 months.

The new data are reported in Extended Data Fig. 3d, and the text was modified accordingly:

Validation of the Long COVID signature in the Zurich cohort.

3. The presence of completely different anti-chemokine antibody signatures for the various comparisons investigated (COVID-19 vs healthy, hospitalisation vs outpatient, long-term symptoms or not) is also puzzling. There is no consistent pathophysiological hypothesis to explain this.

→ We thank the Reviewer for bringing up this point. We would argue that a consistent pathophysiological hypothesis across the 3 signatures is not expected

First, regarding the COVID-19 signature, all COVID-19 patients develop autoantibodies against the same chemokines (CCL19, CCL22 and CXCL17), a finding which is validated in the additional two independent cohorts (see answer to Reviewer 3, points 1 and 2). Consistent with the unique pathophysiology of COVID-19, the pattern of autoantibodies in COVID-19 is different from those in HIV-1, Lyme disease and autoimmunity (see Figure 4 and Extended Data Fig. 9).

Regarding the hospitalization and long COVID signatures, you would not expect that hospitalized and long COVID patients shared the same anti-chemokine antibodies because, as supported by the literature (Mehandru *et al.*, Nat Immunol, 2022; Mantovani *et al.*, Cell Death Differ, 2022; Choutka *et al.*, Nat Med, 2022), both mild and severe COVID-19 individuals can develop long COVID, an observation that is confirmed in all 3 of our cohorts as well. Thus, the data are consistent with different pathophysiologic mechanisms being at play in distinct courses of COVID-19.

4. *The percentage of individuals with persistent symptoms is much higher than in other epidemiological studies. It is unclear which are the clinical criteria on which such persistence was based. There is no consistent diagnosis of long-COVID19 according to the very limited description.*

→ The definition of long COVID in the clinic and in the literature keeps evolving. However, we clearly state in the Methods that in our study long COVID is defined by the persistence of at least one symptom related to COVID-19, and list the clinical features associated with long COVID in Extended Data Fig. 7a,b and Supplementary Table S1. Moreover, the symptoms that we included in the standard questionnaire for assessing long COVID are consistent with those reported by Mehandru *et al.*, Nat Immunol, 2022.

For the original cohort, we cannot exclude that the high frequency (65%) of individuals with long COVID be due to individuals with long COVID being more motivated to enroll and/or being retained in the study. We note, however, that the reported frequency is in line with other publications (*e.g.* Blomberg *et al.*, Nat Med, 2021).

Moreover, although in the Zurich cohort long COVID was defined similarly, the frequency of long COVID was lower (37%). Nevertheless, the overall findings between the two cohorts are consistent with each other.

5. *The authors hypothesise in the Discussion that anti-chemokine antibodies may protect against severe COVID-19 disease. But the major differences were seen here 6 months later, not in the acute phase, how could they have protected?*

→ We thank the Reviewer for raising a valid point. We addressed it directly by measuring anti-chemokine antibodies in the acute phase.

Since the number of available acute samples from the original cohort is small (n=12, see Extended Data Fig. 5d,e and Supplementary Table 1), we obtained acute and 7m samples from a second cohort (Milan, n=44).

Firstly, at 7m, the Milan cohort validates the hospitalization signature obtained with the original cohort. Importantly, the same signature is already present in the acute phase in the Milan cohort and is predictive of disease severity, as shown in Extended Data Fig. 3c:

Validation of the hospitalization signature in the Milan cohort (at t =acute and at t =7months).

The new data and analyses not only validate the original findings at 6-7 months with an independent cohort, but also show that the anti-chemokine antibodies associated with milder disease are induced early after symptoms onset, during the acute phase. Therefore, since they are present during the acute phase, it is reasonable to hypothesize that they may exert a protective effect.

6. *The presence of anti-chemokine antibodies by this methodology does not seem to be specific for COVID-19, being present also in HIV infection. How much is likely that this is important for pathophysiology rather than an epiphenomenon?*

→ We beg to disagree on this point based on the data presented in Figure 4 showing that the pattern of anti-chemokine antibodies in COVID-19 is very different from HIV-1 infection and autoimmune disorders, suggesting a disease-specific role of chemokines and of the respective autoantibodies in pathophysiology.

Consistent with the antigen being required for autoantibodies development, signature chemokines found here to be targeted by autoantibodies are found at high levels in acute COVID-19 (see Figure 3). However, the presence of anti-chemokine antibodies cannot be simply explained as an epiphenomenon (i.e., the more chemokines during infection, the more autoantibodies). For example, for several chemokines that are high in COVID-19 (e.g. CCL3, CCL4, CXCL9), there are no significant autoantibodies developing (but yet they do develop in HIV-1, Fig 4a,b and Extended Data Figs. 6 and 10). This observation alone dismisses a simple “chemokine induces autoantibodies” model. Moreover, there is no significant correlation between chemokine levels and those of the corresponding autoantibodies (see answer to Reviewer 1, Point 6). Finally, plasma chemokines are among the factors most significantly associated with adverse COVID-19 outcome. Instead, the presence of specific cognate autoantibodies is associated with favorable outcome, indicating a more complex relationship, which directly contrasts with the “chemokine induces autoantibodies”, epiphenomenon model.

Reviewer #4

In this study, the authors discovered anti-chemokine Abs were associated with favorable COVID-19 and the lack of long COVID symptoms. In addition, COVID-19 specific anti-chemokine Ab profiles were unique and different HIV and autoimmune diseases. These Abs potentially bind to N-loop of chemokine and could reduce cell migration. In general, these findings are novel but lack of mechanisms.

1. This study found three groups of anti-chemokines Abs which potentially identified COVID-19, hospitalization and long COVID-19. The findings are interesting but lack of mechanisms. Although they perform several blocking assays using anti-chemokine Ab *in vitro*, it is insufficient to address the anti-chemokine Ab function given lack of *in vivo* data.

→ We thank Reviewer 4 for raising the issues of relevance and mechanism.

To address the *in vivo*, physiologic relevance of the findings, we performed new chemotaxis experiments, which demonstrate that plasma IgG from COVID-19 convalescents can indeed significantly inhibit migration of immune cells toward chemokines that attract inflammatory cells such as monocytes and neutrophils, even at concentrations below those present in human blood. This information was added in new Extended Data Fig. 8k, and the Results and Methods modified accordingly:

Polyclonal IgGs from COVID-19 convalescents inhibit chemotaxis.

The new data significantly extend our previous findings because they demonstrate that anti-chemokine antibodies are not only biologically active when present as single monoclonal antibody, but also as polyclonal IgG, which mimics the *in vivo* situation.

In vivo mechanistic studies would require moving from human to animal models. COVID-19 animal models to evaluate disease severity in the post-viral phase (corresponding to the early inflammatory phase associated with hospitalization), as well as models of long COVID, fail to recapitulate the flares of disease severity and long

COVID as they are observed in humans. The non-correspondence between mouse and human chemokines further complicates attempting of *in vivo* studies.

We however provide mechanistic insights by deriving, from COVID-19 convalescents, monoclonal anti-chemokine antibodies against several distinct chemokines, and by demonstrating that they block migration of primary cells through binding to the chemokine N-loop.

2. *What is relationship between the anti-chemokines Ab and chemokines in plasma? The author should detect all of the chemokines and analyzed their levels, and performed the correlations with anti-chemokine Ab levels at early 0.5m, 6m and 12m. These data will provide more useful information of anti-chemokine Abs in clinic.*

→ We agree with Reviewer 4 that this is an important issue and to directly address it, we quantified chemokine concentrations at different time points in two independent cohorts (original and Milan cohorts) and performed correlation analyses with the levels of the corresponding autoantibody in plasma. Several chemokines were significantly elevated during acute disease consistent with the literature (Blanco-Melo *et al.*, Cell, 2020; Liao *et al.*, Nat Med, 2020; Lucas *et al.*, Nature, 2020; Su *et al.*, Cell, 2020; Paludan *et al.*, Sci Immunol, 2022), and some of them remained above baseline levels at 7 months. Of note, none of the chemokines corresponding to autoantibodies of the hospitalization signature were significantly different between mild and severe patients. These new data are presented in new Fig. 3a,b:

Chemokines in plasma during or after COVID-19.

Consistent with this observation, no correlation was observed between levels of chemokines and those of the related signature autoantibodies in the acute phase or at 7 months post infection, as shown in new Fig. 3c:

Lack of correlation between the levels of chemokines and those of the related signature autoantibodies.

Please see also response to Reviewer 1, point 6.

3. *The authors collected samples at 6 month and 12 month and profiled anti-chemokine Abs. They found COVID-19 signatures relation to healthy donor, hospitalization signatures relation to outpatient and more important long COVID-19 signatures. However, COVID-19 and hospitalization signatures were present at 6 months after onset of symptom, which lack of prediction values. The authors should profile early sample to find the COVID-19 and hospitalization signatures because they have collected 0.5m samples.*

4. *Although this study suggests COVID-19 and hospitalization signatures, these anti-chemokine Abs lack of predict and therapeutic values because these signatures happen at 6 months after the onset of symptom rather than early phase of illness. By contrast, long COVID signatures are more important. Alternatively, I suggest Figure 1 and 3 moved into supplemental dataset and manuscript should be reduced into brief report which focus long COVID19 with one main figure.*

Points 3 and 4 are linked since they ask for anti-chemokine antibodies measurements early in the disease (Point 3) and their predictive value (Point 4). This is similar to Point 5 of Reviewer 3.

→ We thank Reviewer 4 for pointing out that the clinical relevance of the findings could be enhanced by data on earlier time points and assessment of their predictive value. To directly address this suggestion, we performed the requested experiments and measured anti-chemokine antibodies in the acute phase.

Since the number of acute samples that are available from the original Lugano cohort is small (n=12), we obtained acute and 7m samples from a second cohort (Milan, n=44).

Firstly, at 7m, the Milan cohort validates the hospitalization signature obtained with the original cohort, as shown in Extended Data Fig. 3c:

Validation of the hospitalization signature in the Milan and Zurich cohorts.

Importantly, the same signatures are already present in the acute phase in the Milan cohort and they accurately distinguish COVID-19 disease (90.5% accuracy) and are predictive of disease severity (85.0% accuracy; Extended Data Fig. 3b,c).

Assignment of individuals to COVID-19 disease (b) or COVID-19 disease severity (c) based on the signatures antibodies by logistic regression analysis in the Milan cohort.

The new data and analyses not only validate the original findings at 6 months with an independent cohort, but also demonstrate that COVID-19 and hospitalization signature anti-chemokine antibodies are induced early upon infection, directly addressing the issue raised by Reviewer 4.

5. *Lack of vaccine incubation information, such as name, incubation time, et al.*

→ For the vaccination cohort, the information is provided in Supplementary Table 1 (Sheet ‘COVID-19 vaccination’): vaccine type, number of doses, number of days between first injection and blood sampling, and number of days between first and second injection. Demographic information on the vaccination cohort is also provided.

Similarly, information about COVID-19 vaccination following natural infection is also provided for the three COVID-19 convalescent cohorts (Lugano, Milan and Zurich; Supplementary Table 1).

Decision Letter, first revision:

Subject: Your manuscript, NI-LE34351A

Message: Our ref: NI-LE34351A

17th Jan 2023

Dear Dr. Robbiani,

Thank you for your patience as we've prepared the guidelines for final submission of your Nature Immunology manuscript, "Anti-chemokine antibodies after SARS-CoV-2 infection correlate with favorable disease course" (NI-LE34351A). Please carefully follow the step-by-step instructions provided in the attached file, and add a response in each row of the table to indicate the changes that you have made. Please also check and comment on any additional marked-up edits we have proposed within the text. Ensuring that each point is addressed will help to ensure that your revised manuscript can be swiftly handed over to our production team.

We would like to start working on your revised paper, with all of the requested files and forms, as soon as possible (preferably by January 24). Please get in contact with us if you anticipate delays.

When you upload your final materials, please include a point-by-point response to any remaining reviewer comments and please make sure to upload your checklist.

If you have not done so already, please alert us to any related manuscripts from your group that are under consideration or in press at other journals, or are being written up for submission to other journals (see: <https://www.nature.com/nature-portfolio/editorial-policies/plagiarism#policy-on-duplicate-publication> for details).

In recognition of the time and expertise our reviewers provide to Nature Immunology's editorial process, we would like to formally acknowledge their contribution to the external peer review of your manuscript entitled "Anti-chemokine antibodies after SARS-CoV-2 infection correlate with favorable disease course". For those reviewers who give their assent, we will be publishing their names alongside the published article.

Nature Immunology offers a Transparent Peer Review option for new original research manuscripts submitted after December 1st, 2019. As part of this initiative, we encourage our authors to support increased transparency into the peer review process by agreeing to have the reviewer comments, author rebuttal letters, and editorial decision letters published as a Supplementary item. When you submit your final files please clearly state in your cover letter whether or not you would like to participate in this initiative. Please note that failure to state your preference will result in delays in accepting your manuscript for publication.

Cover suggestions

As you prepare your final files we encourage you to consider whether you have any images or illustrations that may be appropriate for use on the cover of Nature Immunology.

Nature Immunology has now transitioned to a unified Rights Collection system which will allow our Author Services team to quickly and easily collect the rights and permissions required to publish your work. Approximately 10 days after your paper is formally accepted, you will receive an email in providing you with a link to complete the grant of rights. If your paper is eligible for Open Access, our Author Services team will also be in touch regarding any additional information that may be required to arrange payment for your article.

Please note that *Nature Immunology* is a Transformative Journal (TJ). Authors may publish their research with us through the traditional subscription access route or make their paper immediately open access through payment of an article-processing charge (APC). Authors will not be required to make a final decision about access to their article until it has been accepted. [Find out more about Transformative Journals](https://www.springernature.com/gp/open-research/transformative-journals).

If you have any questions about costs, Open Access requirements, or our legal forms, please contact ASJournals@springernature.com.

Please use the following link for uploading these materials: [REDACTED]

Best regards,

Elle Morris
Senior Editorial Assistant
Nature Immunology
Phone: 212 726 9207
Fax: 212 696 9752
E-mail: immunology@us.nature.com

On behalf of

Ioana Visan, Ph.D.
Senior Editor
Nature Immunology

Tel: 212-726-9207
Fax: 212-696-9752
www.nature.com/ni

Reviewer #1:

Remarks to the Author:

In this manuscript the authors provide a large body of data from multiple independent cohorts showing that COVID-19 is associated with higher levels of specific anti-chemokine autoantibodies. Their claim that COVID-19 has a chemokine auto-antibody profile that is distinct from other infectious and autoimmune diseases is well supported by incorporation of HIV and autoimmune disease datasets. They responded to my specific concerns by accounting for co-variates, measuring the levels of the cognate chemokine for each auto-antibody, and defining capacity for COVID-19 convalescent sera to inhibit immune cell chemotaxis. Please refer to my previous review for a more detailed summary. At this point

I have no additional concerns. The manuscript is clear, well supported by the data and should have high priority for publication.

Reviewer #3:

Remarks to the Author:

The authors responded appropriately to my concerns.

Reviewer #4:

Remarks to the Author:

the authors have addressed the issues well.

Decision Letter, first revision:

Point-by-point response to the Reviewers' comments

Reviewer #1

In this manuscript the authors provide a large body of data from multiple independent cohorts showing that COVID-19 is associated with higher levels of specific anti-chemokine autoantibodies. Their claim that COVID-19 has a chemokine auto-antibody profile that is distinct from other infectious and autoimmune diseases is well supported by incorporation of HIV and autoimmune disease datasets. They responded to my specific concerns by accounting for co-variates, measuring the levels of the cognate chemokine for each auto-antibody, and defining capacity for COVID-19 convalescent sera to inhibit immune cell chemotaxis. Please refer to my previous review for a more detailed summary. At this point I have no additional concerns. The manuscript is clear, well supported by the data and should have high priority for publication.

Reviewer #3

The authors responded appropriately to my concerns.

Reviewer #4

The authors have addressed the issues well.

We thank all the three Reviewers for their effort and time devoted to providing constructive suggestions.

Final Decision Letter:**Subject:** Decision on Nature Immunology submission NI-LE34351B**Message:** In reply please quote: NI-LE34351B

Dear Dr. Robbiani,

I am delighted to accept your manuscript entitled "Autoantibodies against chemokines post-SARS-CoV-2 infection correlate with disease course" for publication in an upcoming issue of Nature Immunology.

Over the next few weeks, your paper will be copyedited to ensure that it conforms to Nature Immunology style. Once your paper is typeset, you will receive an email with a link to choose the appropriate publishing options for your paper and our Author Services team will be in touch regarding any additional information that may be required.

Please note that *Nature Immunology* is a Transformative Journal (TJ). Authors may publish their research with us through the traditional subscription access route or make their paper immediately open access through payment of an article-processing charge (APC). Authors will not be required to make a final decision about access to their article until it has been accepted. [Find out more about Transformative Journals](https://www.springernature.com/gp/open-research/transformative-journals).

Authors may need to take specific actions to achieve [compliance](https://www.springernature.com/gp/open-research/funding/policy-compliance-faqs) with funder and institutional open access mandates. If your research is supported by a funder that requires immediate open access (e.g. according to [Plan S principles](https://www.springernature.com/gp/open-research/plan-s-compliance)) then you should select the gold OA route, and we will direct you to the compliant route where possible. For authors selecting the subscription

publication route, the journal's standard licensing terms will need to be accepted, including [self-archiving policies](https://www.springernature.com/gp/open-research/policies/journal-policies). Those licensing terms will supersede any other terms that the author or any third party may assert apply to any version of the manuscript.

Your paper will be published online soon after we receive your corrections and will appear in print in the next available issue. Content is published online weekly on Mondays and Thursdays, and the embargo is set at 16:00 London time (GMT)/11:00 am US Eastern time (EST) on the day of publication. Now is the time to inform your Public Relations or Press Office about your paper, as they might be interested in promoting its publication. This will allow them time to prepare an accurate and satisfactory press release. Include your manuscript tracking number (NI-LE34351B) and the name of the journal, which they will need when they contact our office.

About one week before your paper is published online, we shall be distributing a press release to news organizations worldwide, which may very well include details of your work. We are happy for your institution or funding agency to prepare its own press release, but it must mention the embargo date and Nature Immunology. Our Press Office will contact you closer to the time of publication, but if you or your Press Office have any enquiries in the meantime, please contact press@nature.com.

Also, if you have any spectacular or outstanding figures or graphics associated with your manuscript - though not necessarily included with your submission - we'd be delighted to consider them as candidates for our cover. Simply send an electronic version (accompanied by a hard copy) to us with a possible cover caption enclosed.

If you have not already done so, we strongly recommend that you upload the step-by-step protocols used in this manuscript to the Protocol Exchange. Protocol Exchange is an open online resource that allows researchers to share their detailed experimental know-how. All uploaded protocols are made freely available, assigned DOIs for ease of citation and fully searchable through nature.com. Protocols can be linked to any publications in which they are used and will be linked to from your article. You can also establish a dedicated page to collect all your lab Protocols. By uploading your Protocols to Protocol Exchange, you are enabling researchers to more readily reproduce or adapt the methodology you use, as well

as increasing the visibility of your protocols and papers. Upload your Protocols at www.nature.com/protocolexchange/. Further information can be found at www.nature.com/protocolexchange/about .

Please note that we encourage the authors to self-archive their manuscript (the accepted version before copy editing) in their institutional repository, and in their funders' archives, six months after publication. Nature Portfolio recognizes the efforts of funding bodies to increase access of the research they fund, and strongly encourages authors to participate in such efforts. For information about our editorial policy, including license agreement and author copyright, please visit www.nature.com/ni/about/ed_policies/index.html

Sincerely,

Ioana Visan, Ph.D.
Senior Editor
Nature Immunology

Tel: 212-726-9207
Fax: 212-696-9752
www.nature.com/ni